# Stable Coresets via Posterior Sampling: Aligning Induced and Full Loss Landscapes

**Wei-Kai Chang**
Purdue University
chang986@purdue.edu

**Rajiv Khanna**
Purdue University
rajivak@purdue.edu

## Abstract

As deep learning models continue to scale, the growing computational demands have amplified the need for effective coreset selection techniques. Coreset selection aims to accelerate training by identifying small, representative subsets of data that approximate the performance of the full dataset. Among various approaches, gradient-based methods stand out due to their strong theoretical underpinnings and practical benefits, particularly under limited data budgets. However, these methods face challenges such as naïve stochastic gradient descent (SGD) acting as a surprisingly strong baseline and the breakdown of representativeness due to loss curvature mismatches over time.

In this work, we propose a novel framework that addresses these limitations. First, we establish a connection between posterior sampling and loss landscapes, enabling robust coreset selection even in high-data-corruption scenarios. Second, we introduce a smoothed loss function based on posterior sampling onto the model weights, enhancing stability and generalization while maintaining computational efficiency. We also present a novel convergence analysis for our sampling-based coreset selection method. Finally, through extensive experiments, we demonstrate how our approach achieves faster training and enhanced generalization across diverse datasets than the current state of the art. (Code are available in: `https://github.com/changwk1001/stable-coreset.git`)

## 1 Introduction

Modern deep learning thrives on massive datasets, but training on all available data can be prohibitively slow. Coreset selection aims to address this issue by selecting a small, representative subset of training data that can be used in lieu of the full dataset, thus accelerating training while preserving model performance. Recent gradient-based coreset methods (Mirzasoleiman et al., 2020; Killamsetty et al., 2021a; Pooladzandi et al., 2022; Yang et al., 2023) have shown promise by approximating the full-data gradient using a subset. In practice, however, we observe a critical failure mode in these approaches: the loss landscape induced by the selected subset is often misaligned with that of the full dataset. This misalignment becomes especially pronounced under realistic conditions like label noise, adversarial corruptions, or very small subset budgets, leading to poor generalization despite the subset's seeming adequacy in matching gradients at selection time.

To illustrate this misalignment problem, consider a model selecting a coreset on a noisy dataset. Gradient-based selection methods tend to prioritize examples with large gradient magnitudes. If some training labels are corrupted (or the model is under-trained, the coreset size is small), such methods can inadvertently overweight outliers or mislabeled examples simply because they induce large immediate loss gradients. The resulting subset produces a distorted loss surface: the model trained on

39th Conference on Neural Information Processing Systems (NeurIPS 2025).

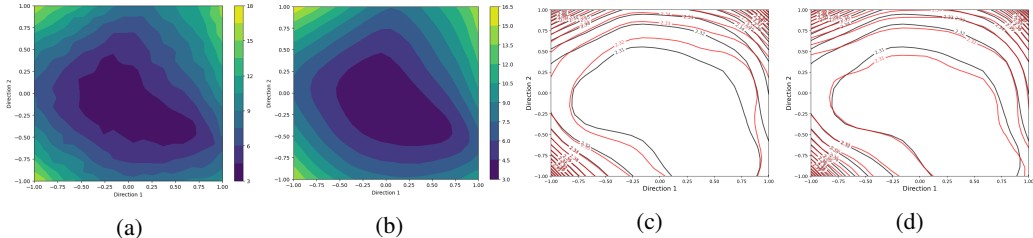

(a)           (b)           (c)           (d)

Figure 1: **(Left) The smoothed surface:** The loss landscape generated by coresets selected by (a) Craig (Mirzasoleiman et al., 2020) The loss landscape generated by the coreset of (b) our method. Our method smoothens the induced landscape. The graph is generated with CIFAR10 data and ResNet20 using 1% data budget. **(Right) Better match in loss landscape:** The black line in both the figures represent the loss landscape of the full training set for MNIST with LeNet using 1% data budget for coreset selection. (d) Our method matches the underlying landscape better than (c) Craig. The mean square error between the two landscapes for Craig and our method are 0.0703 and 0.0662 respectively. (see details in Appendix B.4 and Figure 6.)

this subset will descend along directions that differ from those it would follow if it saw all data (See Section A.8). In other words, the subset's gradient and curvature information (its geometry) deviate from the full data's. In fact, even without noisy labels, minor aberrations in gradient approximations (e.g. due to small coreset bugdet) can distort the induced loss landscape (see Figure 1 (Left)). We term this phenomenon loss landscape misalignment. Over time this discrepancy can magnify – the model's update on the subset can lead it into a region of parameter space that is suboptimal for the true objective. Empirically, we observe that with 20%–50% label noise on benchmarks like CIFAR-10, CIFAR-100, and TinyImageNet, the performance of many gradient-matching based methods degrades catastrophically – in some cases barely above chance – revealing their brittleness in the face of corrupted data.

Prior attempts to address this instability have looked to second-order information. By incorporating Hessian estimates, one can in principle better align a coreset's loss landscape with the full data's landscape. Indeed, recent methods have tried to match not only gradients but also curvatures (e.g. using approximate Hessians) to select more robust subsets (Yang et al., 2023). Unfortunately, these second-order approaches come with serious drawbacks. Computing or approximating Hessians in deep networks is extremely costly and often numerically unstable. The required computations can nullify the very speed-ups that coreset selection aims to achieve (as noted by Okanovic et al. (2023); Mahmood et al. (2025)). Further, we empirically also observe these methods are still susceptible to noise in the labels and can even diverge due to sensitivity to noisy curvature estimates.

We propose a new perspective: posterior smoothing for coreset selection. Rather than deterministically matching gradients or explicitly calculating Hessians, we sample model weight perturbations from a Gaussian posterior centered at the current parameters, and use these perturbed weights to estimate the landscape and evaluate the coreset selection criteria. Intuitively, this strategy smoothens out the loss landscape by marginalizing over a local neighborhood in weight space, effectively simulating a Bayesian posterior over models. By averaging gradient information across multiple weight samples, our method obtains a Monte Carlo smoothing of the coreset objective, and yields a more stable and geometry-aligned selection criterion. This yields a smoothed loss function that closely preserves the underlying structure of the true loss landscape while damping spurious oscillations due to noise or small sample size.

The proposed posterior-sampling strategy is both scalable and theoretically grounded. It requires no explicit Hessian computations; the added overhead is merely a handful of forward passes for sampling, which is much faster compared to full gradient evaluation. We provide a rigorous analysis showing that posterior smoothing provably improves alignment between the induced and full-data loss landscapes. In particular, under standard smoothness assumptions, we prove that our posterior sampling leads to tighter approximation of the true gradient and Hessian of the entire dataset (see Theorem 3.2). This result formalizes the notion of "geometry alignment": the subset sees nearly the same curvature as the full data (see Figure 1 (Right) for a real-world illustration). Finally, we derive convergence guarantees for training on smoothed coresets: under standard assumptions (e.g. smoothness and bounded curvature), we prove that optimizing on the coreset will converge to a

solution that is $\epsilon$-close to the full data optimum. As a side effect, we improve the best known previous rate under the same set of assumptions by $O(1/\sqrt{M})$ for multiplicative noise, where $M$ is the number of samples used in Monte Carlo smoothing.

Empirical results on a wide range of datasets strongly support our claims. Our posterior-based coreset selection consistently outperforms state-of-the-art methods in both accuracy and convergence speed, while using a fraction of the data. The gains are most pronounced in high-noise settings – for instance, with 50% corrupted labels on SNLI, our approach outperforms the next best method by 7% absolute accuracy. Interestingly, for this setting the next best method is Random Sampling — a notoriously strong baseline in data selection. We find that our method handily beats the baselines across almost all settings across multiple datasets (SNLI, TinyImageNet, ImageNet-1k, CIFAR-100, CIFAR-10, MNIST), across different architectures (LeNet on MNIST to ResNet-50 on TinyImageNet/Imagenet-1k and RoBERTa on SNLI), across different subset sizes and different noise corruption levels. Further, our memory footprint is smaller, especially compared to the next most frequently best method (Crest (Yang et al., 2023)) which requires expensive Hessian approximations and sometimes has larger memory footprint (see Figure 2 (Right)) with longer run time than our method and sometimes even full data training. To further solidify the supremacy of our method, we also achieve $20\% - 200\%$ speedup for time-to-highest-accuracy compared to Crest across several different datasets. Our ablation studies reveal practical insights as well. For example, we find that posterior sampling through normalization layers (rather than into all model weights or final-layer weights) provides an optimal trade-off between perturbation and stability, further boosting performance (See Table 2).

To summarize, we make the following technical contributions in this paper:

- **Robust coreset selection via posterior sampling.** We introduce a novel posterior smoothing framework for loss landscape alignment under subset selection. We prove that sampling weights from a local Gaussian posterior yields more faithful gradient and curvature estimates for the subset, resulting in provable improvements in both Hessian alignment and Newton-step similarity between the coreset and full dataset (see Theorem 3.2). This perspective opens up a new family of coreset methods that rigorously account for model uncertainty during selection.

- **Extended convergence theory.** We provide a comprehensive convergence analysis for sampling-based coreset SGD, strengthening and improving prior theoretical results.(see Theorem 3.3) Our analysis accounts for the dual sources of randomness – subset selection and weight sampling – and characterizes how different noise structures (e.g. spherical vs. Hessian-informed Gaussian) influence convergence rates.

- **Exhaustive empirical validation.** Through extensive experiments on vision and NLP benchmarks, we demonstrate that our approach consistently outperforms existing coreset methods across both clean and corrupted datasets across different noise levels and model architectures. Notably, under severe label noise (20–50% of labels corrupted on CIFAR-10, CIFAR-100, TinyImageNet, ImageNet-1k, SNLI), our method remains remarkably robust while prior gradient-based approaches fail. We also highlight key implementation findings – for example, the importance of imposing posteriors in solely the normalization layers – that further enhance coreset effectiveness. Together, our results establish a new state of the art with minimal overhead and memory footprint in both accuracy and stability for coreset selection in deep learning.

For a detailed review of related work, please refer to Appendix A.1.

## 2 Background

The classic training objective is to optimize the Empirical Risk Minimization (ERM) on the training dataset.

$$w^* = \underset{w}{\operatorname{argmin}}\, l(w) = \underset{w}{\operatorname{argmin}} \sum_{i=1}^{n} l_i(w). \tag{1}$$

For tractability and better generalization, in practice, we use SGD (stochastic gradient descent) for optimization of (1) can lead to large memory burden and can be inefficient due to the calculation time.

---

**Algorithm 1** Ensemble Coreset($r, E, T, B, w_i, \eta, P$)

---

1: **Parameters:** Subsample size $R$, Ensemble size $M$, Max epochs $T$, Number of Batch $B$, Initial model parameters $w_0$, Learning rate $\eta$, Number of batch selection $P$, Gaussian Prior $\delta$, minibatch size $m$
2: **for** $t = 1$ to $T$ **do**
3:     **for** $p = 1$ to $P$ **do**
4:         Select random subset $V_p \subseteq S$, $|V_p| = R$
5:         Select $S_p \in \arg\min_{S_p \subseteq V_p} \sum_{i \in V_p} \min_{j \in S_p} E_\delta ||\nabla l_j(w_t + \delta) - \nabla l_i(w_t + \delta)||$, $|S_p| \leq m$
6:     **end for**
6:     $S_t = \bigcup_{p \in [P]}\{S_p\}$
7:     **for** $b = 1$ to $B$ **do**
8:         Sample batch $S_b \subseteq S_t$, $|S_b| = m$
9:         $w_{t,b+1} = w_{t,b} - \eta \nabla l_{S_b}(w_{t,b})$
10:     **end for**
11: **end for**

---

The iterative update of the model parameter can be written as:

$$w_{t+1} = w_t - \eta \nabla l_{s'}(w_t). \tag{2}$$

where $s$ is the random subset sampled from the whole training dataset and the gradient $\nabla l_{s'}(w_t)$ is evaluated on the subset. Several studies have investigated the convergence rate of stochastic gradient descent (SGD) under different settings. In particular, Ghadimi & Lan (2013) established a convergence rate of $\mathcal{O}(\frac{1}{\sqrt{t}})$ , while Xiao et al. (2015) extended this analysis to the mini-batch setting, obtaining $\mathcal{O}(\frac{1}{\sqrt{Rt}})$, where $R$ represents the mini-batch size.

Building up on the SGD, the gradient-based coreset selections (Mirzasoleiman et al., 2020; Killam-setty et al., 2021a; Pooladzandi et al., 2022) aim to find a subset of data whose gradient aligns with the gradient direction of whole training dataset. The formulation is as follows.

$$S^* = \underset{S' \subseteq S, \gamma_j \geq 0}{\arg\min} |S'| \text{ s.t } \max_{w_t \in W} ||\sum_{i \in S} \nabla l_i(w_t) - \sum_{j \in S'} \gamma_j \nabla l_j(w_t)|| \leq \epsilon. \tag{3}$$

Here $\gamma_i$ is the weight for the specific sample $i$. The goal is to jointly optimize for both–the subset $S'$ out of the full dataset $S$ and the weights $\gamma_i$ while ensuring an error of at most $\epsilon$. Practically, the inner optimization over $W$ is often omitted by setting $W = \{w_t\}$, the singleton set of the current iterate in the SGD. As pointed out by Mirzasoleiman et al. (2020), we can transform the problem (3) into a submodular cover problem (see Appendix A.6 for details):

$$S^* = \underset{S' \subseteq S}{\arg\min} |S'| \text{ s.t } \sum_{j \in S'} \min_{i \in S} ||\nabla l_i(w_t) - \nabla l_j(w_t)|| \leq \epsilon. \tag{4}$$

## 3 Posterior-Stable Coreset Selection: New Paradigm for Landscape-Aligned Subsets

The stability in optimization and its importance for generalization is widely studied in the Nguyen et al. (2022); Chen et al. (2018); Lei (2023); Harrison et al. (2022); Attia & Koren (2022). For example, Bisla et al. (2022); Duchi et al. (2012) suggest making the model more stable by sampling model weights with Gaussian posterior. This leads to smoothening of the loss surface and the induced stability can help with generalization. Liu et al. (2022) adapted a similar idea to sharpness-aware minimization algorithm to help stabilize the training process and gain performance advantage.

We build on these previous works and seek a related but different goal. Subset selection can lead to highly unstable and non-smooth loss surfaces (Shin et al., 2023) (see Figure 1). How do we choose a coreset that improves stability of the selection process so that the induced loss landscape matches with the underlying ERM loss landscape ? Towards this end, we first propose a new definition for the stability of the selected coresets:

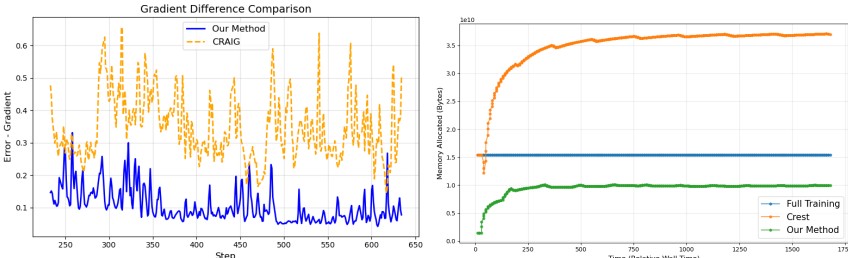

Figure 2: **(Left) Gradient match:** The average gradient estimation error for our method and Craig selection method using LeNet on MNIST. The error is calculated with $\left| \frac{1}{|S|} \sum_{i \in S} \nabla l_i(w_t) - \frac{1}{|S'|} \sum_{j \in S'} \gamma_j \nabla l_j(w_t) \right|$, where $S$ is the training set and $S'$ is the subset selected. Our method generally produces smaller gradient errors and better gradient estimation compared to the Craig method. **(Right) Memory buffer:** The memory consumption for TinyImagenet. Our method shows less memory during the training process. Crest( Yang et al. (2023)) requires to maintain the information of Hessian during training and the intermediate calculation such as checking the threshold calculation of Hessian norm also require large memory buffer. For the plot, we use running average to average over the time step and show the mean value.

**Definition 3.1.** $(\sigma, \epsilon, \bar{w})$-**Stability**. A subset $S'$ is called $(\sigma, \epsilon, \bar{w})$-stable if there exists $\sigma, \epsilon$ such that

$$E_{w \sim N(\bar{w}, \sigma I)} \|\nabla l_{S'}(w) - \nabla l(w)\|_2^2 \le \epsilon, \tag{5}$$

where $\nabla l_{S'}(w)$ is the weighted gradient over the subset $S'$ and $\nabla l(w)$ is the full gradient over the entire dataset. This definition captures the essence of stability by quantifying how much the gradient varies when considering perturbations around the model parameter $\bar{w}$. Similar stability frameworks have been explored in the literature (Bisla et al., 2022; Liu et al., 2022; Duchi et al., 2012) highlighting the crucial role of stability in ensuring that machine learning models for generalization and robustness. A key side-effect of the stability is the smoothening of the function (See Theorem 1 in Bisla et al. (2022)).

**Algorithm** Our goal is to select the $(\sigma, \epsilon, \bar{w})$-stable coresets instead of solving the classic coreset problem (3) by replacing the maximization over the domain $W$ with the stability constraint. Similar to the transformed coreset problem (4), we write our coreset selection optimization problem:

$$S^* = \underset{S' \subseteq S}{\arg\min} |S'| \quad \text{s.t} \quad \sum_{i \in S} \min_{j \in S'} E_\delta \|\nabla l_j(\bar{w} + \delta) - \nabla l_i(\bar{w} + \delta)\| \le \epsilon, \tag{6}$$

where $\delta \sim N(0, \sigma I)$ as $\sigma$ is a hyperparameter or a design variable that controls the perturbation of the weights $\bar{w}$. By optimizing this selection process, we aim to ensure that the coreset $S'$ retains the essential characteristics of the original dataset while adhering to the stability constraints characterized.

Our algorithm is presented in Algorithm 1. (For time complexity analysis, please refer to appendix C) In each epoch $t$, create a pseudo dataset by subsampling stable gradient-matched data points, $P$ times. Instead of solving the constrained selection to match full gradient up to $\epsilon$, we select $m$ data points each time as is standard in other implementations of similar algorithms. This creates a shadow dataset $S_t$ of size $P * m$. We then use the shadow dataset $S_t$ to compute the gradients and optimize the model parameters using batched stochastic gradient descent on $S_t$. This iterative approach allows for refining the model with each epoch, progressively improving its convergence properties and stability. (For more detailed discussion, see Appendix A.7 and Appendix C)

For deep neural networks (DNNs), adding stability to the entire $w_t$ can be prohibitively expensive. Previous works have tried to mitigate this by focusing on the parameters in the last layer of the neural network (Yang et al., 2023; Killamsetty et al., 2021a; Pooladzandi et al., 2022). However, more recent research (Mahmood et al., 2025) has shown that such strategies can be detrimental because of implicit regularization properties of the SGD. Motivated by the recent studies (Mueller et al., 2023; Frankle et al., 2021; Xu et al., 2019) on importance of normalization layers for stability and improved

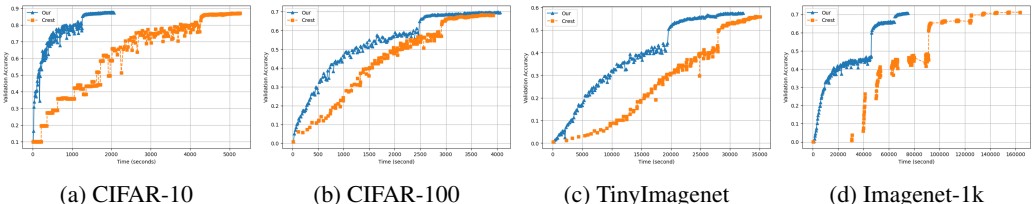

| (a) CIFAR-10 | (b) CIFAR-100 | (c) TinyImagenet | (d) Imagenet-1k |

Figure 3: **Time-to-accuracy:** We plot the time taken to achieve certain validation accuracy against the state-of-the-art Crest Yang et al. (2023) which uses $2^{nd}$ order information for coreset selection and show that our proposed method is more efficient and faster in achieving the same validation accuracies.

predictive performance in DNNs, we focus on our attention for sampling in the batch normalization layers.

We now discuss theoretical properties of stable coresets. Our algorithm can be viewed as coreset selection on the smoothed loss function that leads to stable and better performance on we shall see in the sequel. Furthermore, we establish the relationship between the posterior sampling and the loss landscape. Previous works have proposed that in order to successfully select key samples in training dataset, the loss landscape over selected samples must match to the loss landscape of the whole training dataset. To do so, these works relied on calculating or approximating the parameter Hessian matrix which can be prohibitively expensive and can even offset the speedup obtained by using coresets while also requiring additional memory. Instead, our sampling based strategy is a cheaper and faster way to match the underlying loss landscape without direct calculation Hessians. To understand the relationship between the Gaussian perturbation and loss landscape matching. We propose a our theory in the following. (The proof is in appendix A.2)

**Theorem 3.2.** *Suppose a subset $S' \subset S$ is $(\sigma, \epsilon, w)$-stable and let the Hessian difference be $H_{S',w} - H_{S,w} =: \mathcal{E}$, then,*

*(1) The Hessian difference matrix $\mathcal{E}$ satisfies:*

$$\|\mathcal{E}\| \leq \mathcal{O}(\epsilon^{\frac{1}{2}}) \quad and \quad \mathrm{tr}(\mathcal{E}^2) \leq \mathcal{O}(\frac{\epsilon}{\sigma}).$$

*(2) The difference between newton step of two subset is bounded.*

$$\|H_{S',w}^{-1}\nabla l_{S'}(w) - H_{S,w}^{-1}\nabla l_S(w)\| \leq \mathcal{O}(\epsilon^{\frac{1}{2}}).$$

**Discussion:** Theorem 3.2 formalizes the notion that posterior sampling provides an effective means to ensure the alignment of the loss landscape of the chosen coreset with that of the underlying data. Specifically, if the gradients of a selected subset match those of the full dataset under Gaussian posterior sampling, the discrepancy between their Hessians remains small. This aligns with the findings of Shin et al. (2023), which emphasize the necessity of Hessian similarity across subsets. Furthermore, our approach implicitly ensures that inverse-Hessian weighted gradients are also aligned, satisfying the selection criteria established by Pooladzandi et al. (2022). By leveraging posterior sampling, we circumvent the computational challenges associated with direct Hessian comparisons while preserving theoretical rigor. For even greater control over this alignment, more sophisticated posterior distributions can be designed to precisely fine-tune the loss landscape matching between the coreset and the full training dataset (see appendix A.5).

**Convergence Analysis** Lastly, we also provide convergence analysis for the coresets Mini-batch Stochastic GD along the lines of SGD analysis of Yang et al. (2023) except that we generalize it for the stability setting and improve on their convergence rate.

**Theorem 3.3.** *Say $w \sim N(w_t, \sigma_2 I)$ at epoch $t$. Assume the $l(\cdot)$ is $\beta$-smooth, and the expectation in (6) is calculated by taking $M$ samples. We consider noise in the gradients resulting from the random sampling of batches and coreset selection as $\xi_1$ and $\xi_2$ respectively:*

$$\nabla l(w) = \nabla l_S(w) + \xi_1 + \xi_2$$

*(1) (Absolute noise) If the noise in coreset selection is of the form:*

$$E[\|\xi_2\|] \leq \epsilon,$$

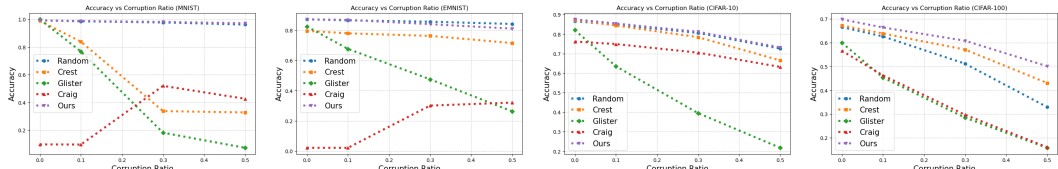

Figure 4: **Strong against all baseline:** Performance at different corrupt ratio with respect to MNIST, EMINST, CIFAR-10 and CIFAR-100. Our method (purple line) consistantly outperform others method at different corrupt ratio, and the performance drop is less when raising the corrupt ratio. Note that the Random sampling method still suffer sharp drop in the high corrupt ratio region in CIFAR-100 dataset.

*then by setting the learning rate to be $\eta = \min\{\frac{1}{\sqrt{T}}, \frac{1}{\beta}\}$ and $\sigma_2^2 d = \frac{1}{M\sqrt{T}}$, We can have convergence rate $\frac{1}{\sqrt{T}}$*

$$E_t ||\nabla l(w_t)||^2 \leq \mathcal{O}(\frac{1}{\sqrt{T}}(l(w_0) - l(w^*) + \frac{\beta^2}{2M} + \frac{\beta\epsilon^2}{M} + \frac{\beta\sigma_1^2}{MR})) \tag{7}$$

*2. (Multiplicative noise) If the noise in estimation of gradients is of the form below*

$$E[||\xi_2||] \leq \epsilon ||\nabla l(w)|| \tag{8}$$

*If we set $\sigma_2^2 d = \frac{1}{M\sqrt{T}}$ and $\eta = \min\{\frac{1}{\sqrt{T}}, \frac{1}{\beta}\}$, we can have convergence rate $\frac{1}{\sqrt{MRT}}$*

$$\frac{1}{T}\sum_{t=0}^{T-1} ||\nabla l(w_t)||^2 = \mathcal{O}(\frac{1}{\sqrt{MRT}}(2(l(w_0) - l(w^*)) + \beta^2 + 2\beta\sigma_1^2)) \tag{9}$$

**Discussion** Theorem 3.3 studies the tradeoff between noise levels and corresponding convergence rates. In case 1 of relatively larger absolute noise, we improve certain terms over previously known standard SGD rates by a factor of $M$, though we have an additional term due to the noise injected into the model weights. These trade-off between different forms of noise actually give us some room for engineering the overall training process. For example, we could avoid the additional term by switching to the naive SGD when close to end of the training. Another observation is that the trade-off will be more favorable towards our method when the random sampling noise $\sigma_1$ becomes large which can happen in the corrupt learning situation (data corruption) making our method more stable in such a situation. In the second case of smaller (proportional to the gradient magnitude) noise, we improve the convergence rate from $\mathcal{O}(\frac{1}{\sqrt{RT}})$ (Yang et al., 2023) to $\mathcal{O}(\frac{1}{\sqrt{MRT}})$.

**Selective Sampling and Stability** Batch Normalization (BN) and related normalization techniques are central to the training and generalization of deep learning models, influencing both the loss landscape and optimization behavior (Santurkar et al., 2019; Sun et al., 2020; Wang et al., 2021). Recent studies also emphasize the role of noise in normalization layers (Kosson et al., 2023; Liang et al., 2019), underscoring their importance in stabilizing learning.

Our method involves sampling around model weights to capture curvature information without explicitly computing the Hessian. However, full-model sampling is computationally expensive, and larger variance (e.g., $\sigma_2^2 d$) leads to smoother loss surfaces but slower convergence (see Theorem 3.3). To balance efficiency and stability, we restrict sampling to the batch normalization layers. This choice is supported by recent findings that highlight the unique role of BN layers in controlling sharpness and enabling efficient optimization (Mueller et al., 2023; Frankle et al., 2021; Xu et al., 2019). While this design is empirically motivated, developing a full theoretical understanding of BN's stabilizing effect remains an open question.

## 4   Experiments: When Does Posterior Sampling Help Most

To evaluate our method. We test our method on ResNet models (ResNet20, ResNet18 and ResNet 50) and transformer based models (ViT, RoBERTa and ELECTRA-small-discriminator) with vision datasets (MNIST, EMNIST, CIFAR10, CIFAR100, TinyImagenet, Imagenet) and language datasets

(SNLI, REC-50). It achieves state-of-the-art accuracy with improved time and memory efficiency, and demonstrates greater robustness to label noise than baselines.[1] All results are averaged over three random seeds for reproducibility. To reduce compute overhead, we apply Gaussian sampling only to batch normalization layers. Each experiment uses 4 sampled models, with $\sigma$ selected via cross-validation from $0.1, 0.01, 0.001$. Sampling is performed dynamically during forward passes, eliminating the need for separate model copies and adding at most one extra layer's memory overhead. This makes our method scalable to larger networks. (See Appendix B for full setup details and Appendix A.9 for more experiment setting such as training budget and archtectures.)

**Gradient Matching: Importance of loss landscape.** As noted by (Yang et al., 2023; Shin et al., 2023), naive optimization of the coreset function (3) to match the gradients without accounting for the loss landscape as done by Craig (Mirzasoleiman et al., 2020) may not actually yield the the best quality subset. We verify this, and show that even on a simple dataset like MNIST, coresets outputted by Craig as vastly inferior to the ones we generate in matching the gradient of the full dataset. The results are presented in Figure 1.

| Dataset | Corruption | Our Method | Random | Crest |
|---------|------------|------------|--------|-------|
| SNLI | 0.0 | **0.9132±0.0013** | 0.9046±0.0020 | 0.9098±0.0022 |
| | 0.1 | **0.8664±0.0012** | 0.8324±0.0054 | 0.8254±0.0028 |
| | 0.3 | **0.7841±0.0021** | 0.7529±0.0031 | 0.7587±0.0042 |
| | 0.5 | **0.6062±0.0016** | 0.5316±0.0024 | 0.5104±0.0055 |
| TinyImageNet | 0.0 | **0.5732±0.0011** | 0.5520±0.0094 | 0.5609±0.0040 |
| | 0.1 | **0.5526±0.0041** | 0.5176±0.0057 | 0.5150±0.0641 |
| | 0.3 | **0.4832±0.0043** | 0.4193±0.0063 | 0.4760±0.0061 |
| | 0.5 | **0.3644±0.0058** | 0.2857±0.0137 | 0.3567±0.0069 |
| ImageNet-1k | 0.0 | 0.7091±0.0004 | 0.7074±0.0004 | **0.7136±0.0015** |
| | 0.1 | **0.6977±0.0016** | 0.6905±0.0016 | 0.6946±0.0035 |
| | 0.3 | **0.6837±0.0007** | 0.6514±0.0001 | 0.6606±0.0024 |
| | 0.5 | **0.6388±0.0008** | 0.5939±0.0017 | 0.6051±0.0009 |

Table 1: **Large scale experiment:** Test accuracy under varying corruption levels on SNLI, TinyImageNet, and ImageNet-1k. Our method consistently outperforms Random and Crest. We did not compare to Craig and Glister on TinyImagent as we run into Out-Of-Memory errors while using code base mentioned in Appendix B. Note that despite Crest marginally outperform our method in Imagenet-1k at zero corruption, the time taken is double of our method (Crest: 42 hours, Ours: 20 hours) along with triple memory consumption.

**The role of sampling in batch normalization.** Our decision to conduct sampling on BN layers is motivated by both computational efficiency and their fundamental role in shaping model behavior. Empirical studies show that sampling on various layers improves performance, but BN layers consistently yield the best results across different models and datasets. Given this trade-off between computational cost and effectiveness, we focus on BN layer sampling to demonstrate the robustness of our method. The table 2 (Left) is performance comparison between sampling at different layers of ResNet20 for CIFAR-10 dataset.

**Test performance: Robustness and Accuracy.** We perform an extensive study across different datasets and label corruption ratios to show the advantage of our method. We summarize the numerical results in Figure 4. For raw numbers, see Table 4 in the appendix. We can observe that our method outperforms established benchmarks across several different settings, especially when the label corruption ratio is high. Most methods suffer a drastic performance drop when the corrupt ratio is increased due to the stability. Naive SGD has been shown to be a strong baseline Okanovic et al. (2023) with little overhead, and our experiments confirm its effectiveness over several traditional baselines. However, its performance also deteriorates significantly with increasing corruption. We attribute this to increased stability of our method compared to other methods. In fact, we also observed that some methods can fail the training as the instability arising out of coreset selections for noisy data can cause the loss to go infinitely large (see caption in Table 4). To further demonstrate the advantage of our method, we conduct an experiment on even higher corrupt ratio (0.6 - 0.9) in table 2 (Right) and find that our method offers strong performance compared to others. Finally, many methods in coreset selection could not scale to large datasets such as SNLI, TinyImagenet and Imagenet-1k (see Table 1).

**Training dynamics: Memory Footprint and Time-to-Accuracy.** As shown in Figure 2, it requires significantly less memory than Crest which performs $2nd$ best accuracy-wise after our method in

---

[1]For the comparison with Pooladzandi et al. (2022), see Appendix E

| Layer | 0.0 | 0.1 | 0.3 | 0.5 |
|---|---|---|---|---|
| All | 0.8654 ± 0.0094 | 0.8479 ± 0.0081 | 0.8079 ± 0.0084 | 0.7288 ± 0.0140 |
| BN | **0.8757 ± 0.0029** | **0.8544 ± 0.0034** | **0.8120 ± 0.0058** | **0.7318 ± 0.0143** |
| FC | 0.8705 ± 0.0019 | 0.8525 ± 0.0033 | 0.8099 ± 0.0043 | 0.7232 ± 0.0171 |

| Dataset | Corrupt Ratio | Our Method | Random | Crest |
|---|---|---|---|---|
| CIFAR-10 | 0.6 | **0.6704 ± 0.0063** | 0.6422 ± 0.0185 | 0.3374 ± 0.0912 |
| | 0.7 | **0.5511 ± 0.0221** | 0.5221 ± 0.0341 | 0.2853 ± 0.0328 |
| | 0.8 | **0.3701 ± 0.0053** | 0.3011 ± 0.0084 | 0.1520 ± 0.0276 |
| | 0.9 | **0.0953 ± 0.0183** | 0.0950 ± 0.0286 | 0.0938 ± 0.0108 |
| CIFAR-100 | 0.6 | **0.3911 ± 0.0063** | 0.2528 ± 0.0131 | 0.3306 ± 0.0054 |
| | 0.7 | **0.2810 ± 0.0048** | 0.1504 ± 0.0036 | 0.2442 ± 0.0028 |
| | 0.8 | **0.1680 ± 0.0107** | 0.0863 ± 0.0211 | 0.1613 ± 0.0084 |
| | 0.9 | **0.0761 ± 0.0029** | 0.0367 ± 0.0087 | 0.0707 ± 0.0061 |

Table 2: **(Left) Sampling at different layers:** CIFAR-10 test accuracy under varying corruption ratios. BN: BatchNorm; FC: Fully Connected; All: All parameters perturbed. **(Right) Robust at even higher corruption:** Performance comparison on CIFAR-10 and CIFAR-100 under corruption ratios from 0.6 to 0.9. Our method consistently outperforms Random selection and Crest.

predictive performance on non-corrupted data,. Crest consumes up to three times more memory than full training due to tracking additional statistics(their algorithm maintains Hessian and accumulated gradient steps for adaptive coreset updates). While their strategy reduces re-selection frequency for coresets, the computational overhead often negates its theoretical potential benefits (Figure 3) and lead to slower training. Our method consistently outperforms Crest across various settings, from small-scale models (MNIST + LeNet) to large-scale architectures (TinyImagenet / Imagenet / SNLI + ResNet50 / RoBERTa).

**Design of more effective posteriors** The choice of posterior can influence the tradeoff between fidelity to the loss landscape and stability. We test with different posteriors - (a) the spherical Gaussian prior with a fixed hyper-parameter $\sigma$, Gaussian with the Hessian-inverse as the covariance matrix, and an Ensemble of models with $4$ different random seed with coresets selected as per Algorithm 1. For Ensemble, we average the gradients of the final layer and use it for coreset calculation. As shown in Table 3, the Spherical-Gaussian posterior achieves the highest final accuracy across various scenarios. We analyze potential failure modes associated with different posteriors, with Figure 5 (in Appendix) illustrating key failure reasons. For CIFAR-10 we observe that Hessian-Gaussian, Ensemble, and Gaussian posteriors require 5962.6s, 3454.2s (parallel), and 2073.5s, respectively, compared to 3553.5s for full training. Notably, only the Spherical-Gaussian posterior offers a speed-up. While training multiple models with different random seeds can be parallelized, the increased memory overhead outweighs the benefits, making it a less favorable trade-off.

| Dataset | Corrupt Ratio | Hessian-Gaussian | Ensemble | Spherical-Gaussian |
|---|---|---|---|---|
| CIFAR-10 | 0.0 | 0.8721 ± 0.0035 | 0.8738 ± 0.0010 | **0.8757 ± 0.0029** |
| | 0.1 | 0.8523 ± 0.0022 | 0.8506 ± 0.0010 | **0.8544 ± 0.0034** |
| | 0.3 | 0.8041 ± 0.0240 | 0.8036 ± 0.0010 | **0.8120 ± 0.0058** |
| | 0.5 | 0.5870 ± 0.0156 | 0.7216 ± 0.0030 | **0.7318 ± 0.0143** |
| CIFAR-100 | 0.0 | 0.6841 ± 0.0045 | 0.6968 ± 0.0070 | **0.6986 ± 0.0025** |
| | 0.1 | 0.6438 ± 0.0043 | 0.6618 ± 0.0030 | **0.6644 ± 0.0034** |
| | 0.3 | 0.5731 ± 0.0032 | 0.6066 ± 0.0020 | **0.6085 ± 0.0044** |
| | 0.5 | 0.4340 ± 0.0051 | 0.4965 ± 0.0060 | **0.5014 ± 0.0068** |

Table 3: **Sampling with different posterior:** Comparison of posterior sampling methods under different corruption ratios. Bold values indicate best performance. We can observe that for method required accurate Hessian estimation, the performance drop sharper compared to the others.

# 5 Future Directions and limitation

The limitation of work lies in that for more general data type like sound or video, the generalization is not guarantee as those data type may impose even harder tasks for the models which is beyond the scope and guarantee in this work.

# Acknowledgments

We thank the Central Indiana Corporate Partnership AnalytiXIN Initiative for their support.

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

# A Appendix

## A.1 Related work

**Coreset methods** To facilitate the training of deep learning models, various methods have been proposed to select informative or representative samples based on the uncertainty of models toward these samples such as Sachdeva et al. (2021) and Coleman et al. (2020). Another line of work selects samples based on their loss difference or degree of error. Methods such as Forgetting Events Toneva et al. (2019), GraNd Paul et al. (2023) have employed this strategy to prioritize samples. These methods aim to reduce the variance of the gradient, thereby improving efficiency of reducing the loss. Another line of work use Bayesian perspective to perform coreset selection Zhang et al. (2021); Guha et al. (2021) which aims to select subset that is equally representative across assigned posteriors.

Additionally, some methods emphasize the centrality of features or embeddings, forming subsets that best represent clusters of samples. Examples include Herding Chen et al. (2012), K-Center Greedy Ding et al. (2019), and Prototypes Sorscher et al. (2023). Another category of methods bases their selection strategy on observations from a validation set Killamsetty et al. (2021b), leveraging additional information to identify the samples most beneficial for training.

Despite lacking rigorous theoretical proof for the advantages of selection strategies, these approaches have demonstrated empirical improvements in speed or performance. Gradient-based methods Mirzasoleiman et al. (2020); Killamsetty et al. (2021a); Pooladzandi et al. (2022) on the other hand, select samples that best approximate the gradient of the entire dataset (training or validation sets). These methods offer theoretically sound support and provide guarantees for the training process. Additionally, several works Zhang et al. (2021); Pooladzandi et al. (2022); Shin et al. (2023) in this direction propose to further match the subset with the loss landscape using information in Hessian matrix and show better convergent result and generalization performance. Our method builds on this direction and addresses the shortcomings of previous work.

**Smoothness and Stability** The optimization of deep learning models has been active area of studied to understand the reasons behind their superior performance in practice. For instance, the Random Weight Perturbation (RWP) algorithm has been shown to smooth the objective function (Bisla et al., 2022; Li et al., 2024) and improve generalization error (Zhou et al., 2019; Jin et al., 2019; Wang & Mao, 2022) despite its simplicity. By perturbing model weights, several studies have demonstrated an increased ability to escape minima with poor generalization and enhance the stability of the optimization process. Our work draws connection to this approach. Instead of explicitly optimizing using a smoothed objective, we select samples that capture similar features, thereby achieving comparable effects. This novel perspective enables us to leverage the benefits of smoothness without directly modifying the optimization objective.

**Bayesian Methods in Deep Learning** Various parameter distributions have been proposed in deep learning to address tasks such as uncertainty estimation, out-of-distribution detection, and classification. Markov chain Monte Carlo (MCMC) methods Chen et al. (2014); Welling & Teh (2011) leverage gradient information for inference, while Laplace approximation-based approaches MacKay (1992); Kirkpatrick et al. (2017); Ritter et al. (2018a) employ Gaussian distributions with the Fisher information matrix or Hessian as the covariance matrix. Other methods explore different strategies: Maddox et al. (2019) average models across different time steps, and Fort et al. (2020) utilize models trained independently with different random seeds. In our work, we evaluate models derived from various posteriors to analyze the trade-offs associated with different sampling strategies.

## A.2 Second order proof

**Theorem A.1.** *Suppose a subset $S' \subset S$ is $\{\sigma, \epsilon, \bar{w}\}$-stable and let the Hessian difference be $H_{S',\bar{w}} - H_{S,\bar{w}} =: \mathcal{E}$, and model has $d$ parameters then,*

*(1) The gradient at difference of subset at $w$ is upper bounded*

$$\|\nabla l_S(\bar{w}) - \nabla l_{S'}(\bar{w})\| \le \frac{1}{2}(c_1\sigma^2 d + \sqrt{c_1^2\sigma^4 d^2 + 4\epsilon}) = \mathcal{O}(\epsilon^{\frac{1}{2}}) \tag{10}$$

*(1) The Hessian difference matrix $\mathcal{E}$ satisfies:*

$$\|\mathcal{E}\| \le c_1\sigma d + \frac{1}{\sigma}\sqrt{c_1^2\sigma^4 d^2 + \sigma(\epsilon - c_2^2\sigma^2 d)} \ \ and \ \ \operatorname{tr}(\mathcal{E}^2) \le \frac{\epsilon - c_2^2\sigma^2 d}{\sigma(1 - 2c_1\sigma d)} \tag{11}$$

*(2) The difference between newton step of two subset is bounded. ($\lambda_{\max}, \lambda_{\min}$) are largest and smallest eigenvalue in $H_{S,w^*}$*

$$\|H_{S',\bar{w}}^{-1}\nabla l_{S'}(\bar{w}) - H_{S,\bar{w}}^{-1}\nabla l_S(\bar{w})\| \leq \frac{1}{\lambda_{\min}}\frac{1}{2}(c_1\sigma^2 d + \sqrt{c_1^2\sigma^4 d^2 + 4\epsilon}) + c\frac{\lambda_{\max}^{\frac{1}{2}}}{\lambda_{\min}^2}(c_1\sigma d +$$

$$\frac{1}{\sigma}\sqrt{c_1^2\sigma^4 d^2 + \sigma(\epsilon - c_2^2\sigma^2 d)}) + \frac{\lambda_{\max}^{\frac{1}{2}}}{\lambda_{\min}^2}\frac{1}{2}(c_1\sigma^2 d + \sqrt{c_1^2\sigma^4 d^2 + 4\epsilon})(c_1\sigma d$$

$$+ \frac{1}{\sigma}\sqrt{c_1^2\sigma^4 d^2 + \sigma(\epsilon - c_2^2\sigma^2 d)}) + \mathcal{O}(\|\mathcal{E}\|^2)$$

*Proof.* Let $f(\bar{w}) = \nabla l_{S'}(\bar{w}) - \nabla l_S(\bar{w})$ and $\nabla f(\bar{w}) = \nabla^2 l_{S'}(\bar{w}) - \nabla^2 l_S(\bar{w})$ (difference in terms of Hessian)

**Assumption 1** Suppose we have the $\nabla f(\bar{w})$ being $2c_1$-Lipschitz Hessian i.e.,
$$\|\nabla f(w) - \nabla f(\bar{w})\| \leq 2c_1\|w - \bar{w}\| \tag{12}$$

**Assumption 2** Suppose we have the $\nabla f(\bar{w})$ being bounded below and above
$$2c_2 I \preceq \nabla f(w), \quad \forall w. \tag{13}$$

**Assumption 3** (Symmetric and non-singular) $\nabla f(w)$ is symmetric and non-singular.

With assumption 1 and assumption 2, we can bound the $f(\bar{w})$ by the following through Hefferon (2017):
$$c_2\|w - \bar{w}\| \leq \|f(w) - f(\bar{w}) - \nabla f(\bar{w})(w - \bar{w})\| \leq c_1\|w - \bar{w}\| \quad \forall w, \bar{w} \tag{14}$$

The assumption is aiming to capture the degree of change of function using polynomial terms. For more discuss about the assumption of the theory, please refer to appendix D

we start with the following

$$\begin{aligned}
\epsilon &\geq \int f(w)^2 p(w) dw \\
&= \int (f(\bar{w}) + f(w) - f(\bar{w}))^2 p(w) dw \\
&= f(\bar{w})^2 + 2f(\bar{w})\int (f(w) - f(\bar{w}))p(w)dw + \int (f(w) - f(\bar{w}))^2 p(w)dw \\
&\geq |f(\bar{w})|^2 - c_1\sigma^2 d|f(\bar{w})| + \int (f(w) - f(\bar{w}))^2 p(w)dw
\end{aligned} \tag{15}$$

For last inequality, we use the following:

$$\begin{aligned}
f(\bar{w})\int (f(w) - f(\bar{w}))p(w)dw &= f(\bar{w})\int (f(w) - f(\bar{w}) - \nabla f(\bar{w})(w - \bar{w}) + \nabla f(\bar{w})(w - \bar{w}))p(w)dw \\
&= f(\bar{w})\int ((f(w) - f(\bar{w}) - \nabla f(\bar{w})(w - \bar{w}))p(w)dw + f(\bar{w})\int \nabla f(\bar{w})(w - \bar{w}))p(w)dw \\
&= f(\bar{w})\int ((f(w) - f(\bar{w}) - \nabla f(\bar{w})(w - \bar{w}))p(w)dw \\
&\leq \|f(\bar{w})\|\int \|(f(w) - f(\bar{w}) - \nabla f(\bar{w})(w - \bar{w})\|p(w)dw \\
&\leq \|f(\bar{w})\|\int c_1\|w - \bar{w}\|p(w)dw \\
&= c_1\|f(\bar{w})\|E_p[\|w - \bar{w}\|] \\
&\leq c_1\|f(\bar{w})\|\sqrt{E_p\|w - \bar{w}\|^2} \\
&= c_1\|f(\bar{w})\|\sigma\sqrt{d}
\end{aligned} \tag{16}$$

We first focus on the first two terms

$$\epsilon \geq \|f(\bar{w})\|^2 - c_1 \sigma^2 d \|f(\bar{w})\| \tag{17}$$

By solving the equation, we can obtain upper bound on the $|f(\bar{w})|$ as following:

$$\|f(\bar{w})\| \leq \frac{1}{2}(c_1 \sigma^2 d + \sqrt{c_1^2 \sigma^4 d^2 + 4\epsilon}) = \mathcal{O}(\epsilon^{\frac{1}{2}}) \tag{18}$$

as a check, if we use linear approximation (i.e., $c_1 = 0$) we will have

$$\|f(\bar{w})\| \leq \sqrt{\epsilon} \tag{19}$$

Now, obtain upper bound on the graident differece at $\bar{w}$, we continue on the cross terms

$$\epsilon \geq \int (f(w) - f(\bar{w}))^2 p(w) dw$$
$$= \int (f(w) - f(\bar{w}) - \nabla f(\bar{w})(w - \bar{w}) + \nabla f(\bar{w})(w - \bar{w}))^2 p(w) dw$$
$$= \int (f(w) - f(\bar{w}) - \nabla f(\bar{w})(w - \bar{w}))^2 p(w) dw$$
$$+2 \int (f(w) - f(\bar{w}) - \nabla f(\bar{w})(w - \bar{w})) \nabla f(\bar{w})(w - \bar{w}) p(w) dw + \int (w - \bar{w}) \nabla f(\bar{w})^2 (w - \bar{w}) p(w) dw \tag{20}$$

we continue by noticing that first term is lower bounded in our assumption.

$$\epsilon \geq c_2^2 \sigma^2 d$$
$$+2 \int (f(w) - f(\bar{w}) - \nabla f(\bar{w})(w - \bar{w})) \nabla f(\bar{w})(w - \bar{w}) p(w) dw + \int (w - \bar{w}) \nabla f(\bar{w})^2 (w - \bar{w}) p(w) dw \tag{21}$$

The first term is obtained through following:

$$\int \|f(w) - f(\bar{w}) - \nabla f(\bar{w})(w - \bar{w})\|^2 p(w) dw \geq \int c_2 \|w - \bar{w}\|^2 p(w) dw$$
$$= c_2 E_p[\|w - \bar{w}\|^2] \tag{22}$$
$$= c_2 \sigma^2 d$$

Now address the cross term,

$$2 \int (f(w) - f(\bar{w}) - \nabla f(\bar{w})(w - \bar{w})) \nabla f(\bar{w})(w - \bar{w}) p(w) dw \geq -2c_1 \int \|w - \bar{w}\|^2 \mathcal{E}(w - \bar{w}) |p(w) dw$$
$$\geq -2c_1 \max_i \lambda_{\mathcal{E},i} \int (w - \bar{w})^2 p(w) dw$$
$$\geq -2c_1 \sigma^2 d \max_i \lambda_{\mathcal{E},i} \tag{23}$$

Now, we address the last term

$$\int (w - \bar{w}) \mathcal{E}^2 (w - \bar{w}) 2\pi^{-\frac{d}{2}} \det(\sigma I)^{-\frac{1}{2}} \exp(-\frac{1}{2}(w - \bar{w})(\sigma I)^{-1}(w - \bar{w})) dw \tag{24}$$

By change of variable $u = \nabla f(\bar{w})(w - \bar{w})$, we can obtain the following by assuming that $\mathcal{E}$ is not degenerate:

$$
\begin{aligned}
\int &|\det(\mathcal{E}^{-1})| u^T u 2\pi^{-\frac{d}{2}} \det(\sigma I)^{-\frac{1}{2}} \exp(-\frac{1}{2} u^T \mathcal{E}_1^{-1} (\sigma I)^{-1} \mathcal{E}^{-1} u) du \\
&= |\det(\mathcal{E}^{-1})| \frac{\det(\mathcal{E}(\sigma I)\mathcal{E})^{\frac{1}{2}}}{\det((\sigma I))^{\frac{1}{2}}} \operatorname{tr}(\mathcal{E}(\sigma I)\mathcal{E}) \\
&= \operatorname{tr}(\sigma I \mathcal{E}^2) \\
&\geq \sigma \operatorname{tr}(\mathcal{E}^2) \\
&\geq \sigma \max_i \lambda_{\mathcal{E},i}^2
\end{aligned}
\tag{25}
$$

We assume the $H^{-1}$ and $\nabla f(\bar{w})$ are symmetry matrix. Here, we utilize two identities for the first equility and second inequility:

$$
\operatorname{tr}(ABC) = \operatorname{tr}(BCA)
\tag{26}
$$

The second identity follows the following derivation by first noting that symmetry matrix can be decompose into the spectral form $A = \sum_i \lambda_{A,i} v_i v_i^T$

$$
\begin{aligned}
\operatorname{tr}(AB) &= \sum_i \lambda_{B,i}(v_i^T A v_i) \\
&\geq \min_i \lambda_{B,i} \sum_i (v_i^T A v_i) \\
&= \min_i \lambda_{B,i} \operatorname{tr}(A)
\end{aligned}
\tag{27}
$$

Now, we put everything together,

$$
\epsilon \geq c_2^2 \sigma^2 d - 2c_1 \sigma^2 d \max_i \lambda_{\mathcal{E},i} + \sigma \max_i \lambda_{\mathcal{E},i}^2
\tag{28}
$$

we can solve for the largest difference in eigenvalue

$$
c_1 \sigma d + \frac{1}{\sigma}\sqrt{c_1^2 \sigma^4 d^2 + \sigma(\epsilon - c_2^2 \sigma^2 d)} \geq \max_i \lambda_{\mathcal{E},i}
\tag{29}
$$

Therefore, we can have that

$$
\|H_{S',\bar{w}} - H_{S,\bar{w}}\| \leq \mathcal{O}(\epsilon^{\frac{1}{2}})
\tag{30}
$$

We can also obtain upper bound on the difference in Hessian in terms of trace

$$
\begin{aligned}
\epsilon - c_2^2 \sigma^2 d &\geq -2c_1 \sigma^2 d \max_i \lambda_{\mathcal{E}_1,i} + \sigma \max_i \lambda_{\mathcal{E}_1,i}^2 \\
&\geq -2c_1 \sigma^2 d \operatorname{tr}(\mathcal{E}^2) + \sigma \operatorname{tr}(\mathcal{E}^2) \\
&\geq \sigma(1 - 2c_1 \sigma d) \operatorname{tr}(\mathcal{E}^2)
\end{aligned}
\tag{31}
$$

and therefore

$$
\frac{\epsilon - c_2^2 \sigma^2 d}{\sigma(1 - 2c_1 \sigma d)} \geq \operatorname{tr}(\mathcal{E}^2)
\tag{32}
$$

Now, to prove the newton step is also similar, we write the overall into the following equation

$$
\|H_{S',\bar{w}}^{-1} \nabla'_s l(\bar{w}) - H_{S,\bar{w}}^{-1} \nabla l_S(\bar{w})\| = \|(H_{S,\bar{w}} + \mathcal{E})^{-1}(\nabla l_S(\bar{w}) + \mathcal{E}_2) - H_{S,\bar{w}}^{-1} \nabla l_S(\bar{w})\|
\tag{33}
$$

To obtain the upper bound of the Hessian inverse, we use inverse perturbation theory

$$(H_{S,\bar{w}} + \mathcal{E})^{-1} = H_{S,\bar{w}}^{-1} - H_{S,\bar{w}}^{-1}\mathcal{E}H_{S,\bar{w}}^{-1} + \mathcal{O}(||\mathcal{E}||^2) \tag{34}$$

Now, we obtain upper bound on the magnitude of eigenvalue of the difference on the Hessian for the subset.(i.e., $\max_i |\lambda_{\mathcal{E},i}| \leq c_1\sigma d + \frac{1}{\sigma}\sqrt{c_1^2\sigma^4 d^2 + \sigma(\epsilon - c_2^2\sigma^2 d)}$) and the gradient difference $||\nabla l_{s'}(w) - \nabla l_s(w)|| \leq \frac{1}{2}(c_1\sigma^2 d + \sqrt{c_1^2\sigma^4 d^2 + 4\epsilon})$. We can follow to upper bound the desired quantity $||H_{S',\bar{w}}^{-1}\nabla l_{S'}(w) - H_{S,\bar{w}}^{-1}\nabla l_s(w)||_2^2$. At this point, we assume that the gradient has bounded magnitude $||\nabla l_S(w)|| \leq c, \ \forall S, w$

Put in the above, and assume that the gradient has bounded magnitude $||\nabla l_S(w)|| \leq c \ \forall S, w$

$$||H_{S',\bar{w}}^{-1}\nabla'_s l(\bar{w}) - H_{S,\bar{w}}^{-1}\nabla l_s(\bar{w})||$$
$$= ||(H_{S,\bar{w}} + \mathcal{E})^{-1}(\nabla l_S(\bar{w}) + \mathcal{E}_2) - H_{S,\bar{w}}^{-1}\nabla l_s(\bar{w})||$$
$$\leq ||H_{S,\bar{w}}^{-1}\mathcal{E}_2 - H_{S,\bar{w}}^{-1}\mathcal{E}H_{S,\bar{w}}^{-1}\nabla l_S(\bar{w}) + H_{S,\bar{w}}^{-1}\mathcal{E}H_{S,\bar{w}}^{-1}\mathcal{E}_2) + \mathcal{O}(||\mathcal{E}||^2)||$$
$$\leq ||H_{S,\bar{w}}^{-1}\mathcal{E}_2|| + ||H_{S,\bar{w}}^{-1}\mathcal{E}H_{S,\bar{w}}^{-1}\nabla l_S(\bar{w})|| + ||H_{S,\bar{w}}^{-1}\mathcal{E}H_{S,\bar{w}}^{-1}\mathcal{E}_2)|| + \mathcal{O}(||\mathcal{E}||^2)$$
$$\leq \frac{1}{\lambda_{\min}}\frac{1}{2}(c_1\sigma^2 d + \sqrt{c_1^2\sigma^4 d^2 + 4\epsilon}) + c\frac{\lambda_{\max}^{\frac{1}{2}}}{\lambda_{\min}^2}(c_1\sigma d + \frac{1}{\sigma}\sqrt{c_1^2\sigma^4 d^2 + \sigma(\epsilon - c_2^2\sigma^2 d)}) + \frac{\lambda_{\max}^{\frac{1}{2}}}{\lambda_{\min}^2}\frac{1}{2}(c_1\sigma^2 d + \sqrt{c_1^2\sigma^4 d^2 + 4\epsilon})(c_1\sigma d$$
$$+ \frac{1}{\sigma}\sqrt{c_1^2\sigma^4 d^2 + \sigma(\epsilon - c_2^2\sigma^2 d)}) + \mathcal{O}(||\mathcal{E}||^2)$$
$$\leq \mathcal{O}(\epsilon^{\frac{1}{2}})$$
$$\tag{35}$$

At this point, we can conclude that

$$||H_{S',\bar{w}}^{-1}\nabla'_s l(\bar{w}) - H_{S,\bar{w}}^{-1}\nabla l_s(\bar{w})||_2^2 \leq \mathcal{O}(\epsilon) \tag{36}$$

$\square$

### A.3 Proof for Theorem 3.3

Next, the assumptions required for the proof are listed below:
**Assumption 1**. Bounded variance and unbiased estimator of the random sampling and coreset selection subrutine:

$$E[||\nabla l_S(w) - \nabla l(w)||^2] \leq \sigma_1^2 \tag{37}$$
$$E[\nabla l_S(w)] = \nabla l(w) \tag{38}$$

**Assumption 2**. $\alpha$-lipschitz continuity:

$$l(w) - l(v) \leq \alpha ||w - v||_2 \tag{39}$$

**Assumption 3**. $\beta$-smoothness:

$$||\nabla l(w) - \nabla l(v)|| \leq \beta ||w - v||_2 \tag{40}$$

**Assumption 4**. $\epsilon_i \in \Re^d$ sampled i.i.d from diagonal gaussian:

$$\epsilon_i \sim N(0, \sigma_2 I), \ \ \forall i = 1...M \tag{41}$$

There exist two randomnesses in our formulation. One is the subset obtained through random sampling from the whole dataset. The other randomness results from the noise inject to model weight.

$$l(w_{t+1}) \leq l(w_t) + \nabla l(w_t)^T(w_{t+1} - w_t) + \frac{\beta}{2}||w_{t+1} - w_t||^2$$
$$\leq l(w_t) - \eta_t\nabla l(w_t)^T\frac{1}{M}\sum_{i=1}^{M}\nabla_s l(w_t + \epsilon_i) + \frac{\beta\eta_t^2}{2}||\frac{1}{M}\sum_{i=1}^{M}\nabla_S l(w_t + \epsilon_i)||^2 \tag{42}$$

We rewrite the last term as follows:

$$\frac{\beta\eta_t^2}{2}||\frac{1}{M}\sum_{i=1}^{M}\nabla_S l(w_t+\epsilon_i)||^2 = \frac{\beta\eta_t^2}{2}(||\frac{1}{M}\sum_{i=1}^{M}\nabla_S l(w_t+\epsilon_i)-\nabla l(w_t)||^2 + \frac{2}{M}\sum_{i=1}^{M}\nabla_S l(w_t+\epsilon_i)\nabla l(w_t) - ||\nabla l(w_t)||^2)$$

(43)

Input the above into the original formulation:

$$l(w_{t+1}) \leq l(w_t) - \eta_t \nabla l(w_t)^T \sum_{i=1}^{M}\nabla_S l(w_t+\epsilon_i) + \frac{\beta\eta_t^2}{2}||\frac{1}{M}\sum_{i=1}^{M}\nabla_S l(w_t+\epsilon_i)||^2$$

$$\leq l(w_t) - \eta_t(1-\eta_t\beta)\nabla l(w_t)^T \frac{1}{M}\sum_{i=1}^{M}\nabla_S l(w_t+\epsilon_i) + \frac{\beta\eta_t^2}{2}(||\frac{1}{M}\sum_{i=1}^{M}\nabla_S l(w_t+\epsilon_i)-\nabla l(w_t)||^2 - ||\nabla l(w_t)||^2)$$

$$\leq l(w_t) - \eta_t(1-\eta_t\beta)\nabla l(w_t)^T \frac{1}{M}\sum_{i=1}^{M}\nabla_S l(w_t+\epsilon_i) - \frac{\beta\eta_t^2}{2}||\nabla l(w_t)||^2 +$$

$$\beta\eta_t^2||\frac{1}{M}\sum_{i=1}^{M}\nabla_S l(w_t+\epsilon_i)-\nabla l(w_t+\epsilon_i)||^2 + \beta\eta_t^2||\frac{1}{M}\sum_{i=1}^{M}\nabla l(w_t+\epsilon_i)-\nabla l(w_t)||^2$$

(44)

The last two terms are achieved through the identity $||a-b|| \leq 2(||a-c||+||b-c||)$. We now want to resolve the second term in the formulation.

$$\nabla l(w_t)^T \frac{1}{M}\sum_{i=1}^{M}\nabla_S l(w_t+\epsilon_i) = \nabla l(w_t)^T (\frac{1}{M}\sum_{i=1}^{M}\nabla_S l(w_t+\epsilon_i) - \nabla l(w_t) + \nabla l(w_t))$$

$$= ||\nabla l(w_t)||^2 + \nabla l(w_t)^T (\frac{1}{M}\sum_{i=1}^{M}\nabla_S l(w_t+\epsilon_i) - \nabla l(w_t))$$

$$= ||\nabla l(w_t)||^2 +$$

$$\nabla l(w_t)^T (\frac{1}{M}\sum_{i=1}^{M}(\nabla_S l(w_t+\epsilon_i) - \nabla l(w_t+\epsilon_i)) + \frac{1}{M}\sum_{i=1}^{M}\nabla l(w_t+\epsilon_i) - \nabla l(w_t))$$

(45)

The term can be simplified by taking the expectation over the randomness in a stochastic subset.

$$E_S[\nabla l(w_t)^T \frac{1}{M}\sum_{i=1}^{M}\nabla_S l(w_t+\epsilon_i)] = ||\nabla l(w_t)||^2 + \nabla l(w_t)^T (\frac{1}{M}\sum_{i=1}^{M}\nabla l(w_t+\epsilon_i) - \nabla l(w_t))$$

$$\leq ||\nabla l(w_t)||^2 + \frac{1}{2}||\nabla l(w_t)||^2 + \frac{1}{2}(||\frac{1}{M}\sum_{i=1}^{M}\nabla l(w_t+\epsilon_i) - \nabla l(w_t)||^2)$$

(46)

The last term is achieved through the Cauchy-Schwarz inequality. Therefore,

$$-E_s[\nabla l(w_t)^T \frac{1}{M}\sum_{i=1}^{M}\nabla_S l(w_t+\epsilon_i)] \leq -\frac{1}{2}||\nabla l(w_t)||^2 + \frac{1}{2}(||\frac{1}{M}\sum_{i=1}^{M}\nabla l(w_t+\epsilon_i) - \nabla l(w_t)||^2)$$

(47)

We further simplified the formulation by taking the expectation over the randomness in noise injected to the model weight

$$-E_{s,\epsilon_i,i=1...d}[\nabla l(w_t)^T \frac{1}{M}\sum_{i=1}^{M}\nabla_S l(w_t+\epsilon_i)] \leq -\frac{1}{2}||\nabla l(w_t)||^2 + \frac{1}{2}\beta^2\sigma_2^2 d \qquad (48)$$

Next, we first solve the fifth term in the original formulation taking the expectation with respect to $\epsilon_i$ for each $i$.

$$\beta\eta_t^2 E[||\frac{1}{M}\sum_{i=1}^{M}\nabla l(w_t+\epsilon_i) - \nabla l(w_t)||^2] \leq \beta\eta_t^2 E[||\frac{1}{M}\sum_{i=1}^{M}\beta\epsilon_i||^2]$$

$$\leq \beta^3\eta_t^2\frac{1}{M}\sum_{i=1}^{M}E[||\epsilon_i||^2] \tag{49}$$

$$\leq \beta^3\eta_t^2\sigma_2^2 d$$

Finally, we deal with the term.

$$\beta\eta_t^2||\frac{1}{M}\sum_{i=1}^{M}\nabla_S l(w_t+\epsilon_i) - \nabla l(w_t+\epsilon_i)||^2 \tag{50}$$

Here, we assume two different errors that can arise in practice. The first is the random sampling batch created by sampling from the entire dataset. The second error originates from the coreset approximation. We formulate as follows:

$$\nabla l(w) = \nabla_S l(w) + \xi_1 + \xi_2 \tag{51}$$

The $\xi_1$ is the result of the stochasticity of the random batch generation. The $\xi_2$ is the error that originates from the selection of the coreset. We consider two different forms of error in $\xi_2$. One is the absolute error and the other is that the error is propotional to the gradient. We formulate as follows:

$$E[||\xi_2||] \leq \epsilon \quad \text{or} \quad E[||\xi_2||] \leq \epsilon||\nabla l(w)||, \quad \epsilon > 0 \tag{52}$$

Note: we assume here that these two errors $\xi_1, \xi_2$ are independent to each other.

**Situation 1** We first consider the case where the error is absolute.

$$\beta\eta_t^2 E_{S,\epsilon_i,i=1...M} E||\frac{1}{M}\sum_{i=1}^{M}\nabla_S l(w_t+\epsilon_i) - \nabla l(w_t+\epsilon_i)||^2 \leq \beta\eta_t^2\frac{1}{M^2}\sum_{i=1}^{M}E||\nabla_S l(w_t+\epsilon_i) - \nabla l(w_t+\epsilon_i)||^2$$

$$= \beta\eta_t^2\frac{1}{M^2}\sum_{i=1}^{M}E[||\xi_{1,i}||^2 + 2\xi_{1,i}\xi_{2,i} + ||\xi_{2,i}||^2]$$

$$= \beta\eta_t^2\frac{1}{M}(||\xi_1||^2 + 2\xi_1\xi_2 + ||\xi_2||^2)$$

$$\leq \beta\eta_t^2\frac{1}{M}(\frac{\sigma_1^2}{R} + \epsilon^2) \tag{53}$$

The R is the batch size for each batch of random sampling. The cross terms are eliminated due to the assumption 5.

Integrate those terms into the original formulation.

$$l(w_{t+1}) \leq l(w_t) - \eta_t||\nabla l(w_t)||^2 + \frac{\eta_t(1-\eta_t\beta)}{2}\beta^2\sigma_2^2 d + \frac{\beta\eta_t^2\sigma_1^2}{MR} + \frac{\beta\eta_t^2\epsilon^2}{M} + \beta^3\eta_t^2\sigma_2^2 d \tag{54}$$

Here, we pick $\eta_t = \eta$ (fixed step size) and $\eta \leq \frac{1}{\beta}$. Rearrange and sum over time step, and we will have as follows:

$$\eta\sum_{t=0}^{T-1}||\nabla l(w_t)||^2 \leq l(w_0) - l(w^*) + \frac{\beta^2\sigma_2^2 d}{2}\sum_{t=0}^{T-1}\eta(1+\eta\beta) + \frac{\beta\epsilon^2}{M}\sum_{t=0}^{T-1}\eta^2 + \frac{\beta\sigma_1^2}{MR}\sum_{t=0}^{T-1}\eta^2 \tag{55}$$

Here, we divide on both sides by $T\eta$

$$
\begin{aligned}
\frac{1}{T}\sum_{t=0}^{T-1}||\nabla l(w_t)||^2 &\le \frac{1}{T\eta}(l(w_0)-l(w^*)) + \frac{\beta^2\sigma_2^2 d}{2T}\sum_{t=0}^{T-1}(1+\eta\beta) + \frac{\beta\epsilon^2}{M}\eta + \frac{\beta\sigma_1^2}{MR}\eta \\
&= \frac{1}{T\eta}(l(w_0)-l(w^*)) + \frac{\beta^2\sigma_2^2 d}{2} + \frac{\beta^3\sigma_2^2 d}{2}\eta + \frac{\beta\epsilon^2}{M}\eta + \frac{\beta\sigma_1^2}{MR}\eta
\end{aligned}
\tag{56}
$$

As we have control for both $\eta$ and $\sigma_2^2 d$, we pick $\sigma_2^2 d = \frac{1}{M\sqrt{T}}$ and $\eta = \min\{\frac{1}{\sqrt{T}},\frac{1}{\beta}\}$ and, therefore,

$$
\frac{1}{T}\sum_{t=0}^{T-1}||\nabla l(w_t)||^2 \le \frac{1}{T}(l(w_0)-l(w^*))\max\{\sqrt{T},\beta\} + \frac{1}{\sqrt{T}}(\frac{\beta^2}{2M}+\frac{\beta\epsilon^2}{M}+\frac{\beta\sigma_1^2}{MR}) + \frac{1}{T}\frac{\beta^3}{2M}
\tag{57}
$$

If we stop at any specific time step with probability $\frac{1}{T}$, and we observe that the average gradient exist convergent rate $\frac{1}{\sqrt{T}}$ for $T$ large enough which is:

$$
\begin{aligned}
E_t||\nabla l(w_t)||^2 &\le \frac{1}{\sqrt{T}}(l(w_0)-l(w^*)) + \frac{1}{\sqrt{T}}(\frac{\beta^2}{2M}+\frac{\beta\epsilon^2}{M}+\frac{\beta\sigma_1^2}{MR}) + \frac{1}{T}\frac{\beta^3}{2M} \\
&= \mathcal{O}(\frac{1}{\sqrt{T}}(l(w_0)-l(w^*) + \frac{\beta^2}{2M}+\frac{\beta\epsilon^2}{M}+\frac{\beta\sigma_1^2}{MR}))
\end{aligned}
\tag{58}
$$

**Situation 2** We now consider the case where the error is propotional to the magnitude of the gradient. i.e.,

$$
E[||\xi_2||] \le \epsilon||\nabla l(w)||, \quad \epsilon > 0
\tag{59}
$$

We analyze the term as follows:

$$
\begin{aligned}
\beta\eta_t^2 E[||\frac{1}{M}\sum_{i=1}^{M}\nabla_S l(w_t+\epsilon_i) - \nabla l(w_t+\epsilon_i)||^2] &\le \beta\eta_t^2\frac{1}{M}E[||\xi_1||^2 + 2\xi_1\xi_2 + ||\xi_2||^2] \\
&\le \beta\eta_t^2\frac{1}{M}(\frac{\sigma_1^2}{R}+\epsilon^2||\nabla l(w_t)||^2)
\end{aligned}
\tag{60}
$$

Integrate the term to previous result.

$$
l(w_{t+1}) \le l(w_t) - (\eta_t - \frac{\beta\eta_t^2\epsilon^2}{M})||\nabla l(w_t)||^2 + \frac{\eta_t(1-\eta_t\beta)}{2}\beta^2\sigma_2^2 d + \frac{\beta\eta_t^2\sigma_1^2}{MR} + \beta^3\eta_t^2\sigma_2^2 d
\tag{61}
$$

Set $\eta_t = \eta$. We rearrange and perform same operation and we get:

$$
\begin{aligned}
\sum_{t=0}^{T-1}(\eta - \frac{\beta\eta^2\epsilon^2}{M})||\nabla l(w_t)||^2 &\le l(w_0)-l(w^*) + \frac{\beta^2\sigma_2^2 d}{2}\sum_{t=0}^{T-1}\eta(1+\eta\beta) + \frac{\beta\sigma_1^2}{MR}\sum_{t=0}^{T-1}\eta^2 \\
&= l(w_0)-l(w^*) + \frac{\beta^2\sigma_2^2 d}{2}T\eta(1+\eta\beta) + \frac{\beta\sigma_1^2}{MR}T\eta^2
\end{aligned}
\tag{62}
$$

Divide by $T(\eta - \frac{\beta\eta^2\epsilon^2}{M})$ on both sides and choose step size such that $(1-\frac{\beta^2\eta\epsilon^2}{M}) \ge \frac{1}{2}$

$$
\frac{1}{T}\sum_{t=0}^{T-1}||\nabla l(w_t)||^2 \le \frac{2}{T}(l(w_0)-l(w^*)) + \beta^2\sigma_2^2 d(1+\eta\beta) + \frac{2\beta\sigma_1^2}{MR}\eta
\tag{63}
$$

We pick $\sigma_2^2 d = \frac{1}{\sqrt{MRT}}$ and $\eta = \frac{\sqrt{MR}}{\sqrt{T}}$ and we will have

$$\frac{1}{T}\sum_{t=0}^{T-1}||\nabla l(w_t)||^2 \leq \frac{2}{T}(l(w_0) - l(w^*)) + \frac{\beta^2}{\sqrt{MRT}}(1 + \beta\frac{\sqrt{MR}}{\sqrt{T}}) + \frac{2\beta\sigma_1^2}{\sqrt{MRT}}$$

$$= \mathcal{O}(\frac{1}{\sqrt{MRT}}(2(l(w_0) - l(w^*)) + \beta^2 + 2\beta\sigma_1^2)) \tag{64}$$

Similar to the first situation, we will have convergence rate with $\frac{1}{\sqrt{MRT}}$ which is $\frac{1}{\sqrt{M}}$ faster than the naive SGD.

## A.4 Data summary

| Dataset | Corrupt Ratio | Random | Crest | Glister | Craig | Ours |
|---|---|---|---|---|---|---|
| MNIST | 0 | **0.9921±0.0008** | 0.9892±0.0005 | 0.9997±0.0001 | (*)0.0972±0.0 | 0.9914±0.0003 |
| | 0.1 | 0.9854±0.0009 | 0.8384±0.0812 | 0.7676±0.0271 | 0.0961±0.0009 | **0.9876±0.0009** |
| | 0.3 | 0.9772±0.0008 | 0.3374±0.3385 | 0.1815±0.0664 | 0.5196±0.2346 | **0.9809±0.0025** |
| | 0.5 | 0.962±0.0019 | 0.3277±0.2062 | 0.0725±0.0109 | 0.4264±0.0131 | **0.9715±0.0021** |
| EMNIST | 0 | 0.8713±0.0026 | 0.7955±0.0196 | 0.8246±0.0049 | (*)0.0219±0.0 | **0.8715±0.0015** |
| | 0.1 | 0.8649±0.0016 | 0.7788±0.0132 | 0.6762±0.0094 | 0.0218±0.0007 | **0.8679±0.0012** |
| | 0.3 | **0.8557±0.0007** | 0.7621±0.0044 | 0.4748±0.0236 | 0.301±0.2553 | 0.8396±0.0016 |
| | 0.5 | **0.8409±0.0024** | 0.7146±0.0051 | 0.2627±0.0172 | 0.321±0.0111 | 0.8100±0.0029 |
| CIFAR-10 | 0 | 0.8660±0.0015 | 0.8724±0.004 | 0.8207±0.0097 | 0.7618±0.008 | **0.8757±0.0029** |
| | 0.1 | 0.8481±0.001 | 0.8440±0.003 | 0.6350±0.0261 | 0.7490±0.0066 | **0.8544±0.0034** |
| | 0.3 | 0.8063±0.010 | 0.7843±0.004 | 0.3953±0.0548 | 0.7050±0.0112 | **0.8120±0.0058** |
| | 0.5 | 0.7257±0.015 | 0.6652±0.013 | 0.2175±0.0292 | 0.6320±0.0186 | **0.7318±0.0143** |
| CIFAR-100 | 0 | 0.6659±0.0041 | 0.6728±0.003 | 0.6004±0.0052 | 0.5665±0.0053 | **0.6986±0.0025** |
| | 0.1 | 0.6268±0.005 | 0.6391±0.004 | 0.4538±0.0076 | 0.4629±0.0102 | **0.6644±0.0034** |
| | 0.3 | 0.5119±0.014 | 0.5712±0.006 | 0.2846±0.0074 | 0.2968±0.0162 | **0.6085±0.0044** |
| | 0.5 | 0.3293±0.014 | 0.4305±0.026 | 0.1578±0.0218 | 0.1603±0.0048 | **0.5014±0.0068** |

Table 4: Performance comparison of different methods across datasets and corruption ratios. Results are reported as mean ± standard deviation. Each experiments are average over 5 different random seed. The best performance for each setting is highlighted in bold. The * mark the situation in which has diverging behavior during the optimization. The situation usually occurs in Craig method for high coruption scenario.

## A.5 Discussion about different posterior

**Hessian inverse covariance** is the one containing exact information about loss landscape of the models at certain training points and it is also the posterior used in deriving the theory for sampling. However, calculation of Hessian can introduce large memory footprint as it will require us to keep track of the Hessian during training as listed in works Yang et al. (2023). Not only it can cause large memory footprint, it is also hard to calculate and require steps of approximation. The selection strategy involving calculation of Hessian will inevitably be slowed down as it requires at least one forward, backward propogation and the intermediate calculation. Another issue about Hessian inverse is that despite it contains the curvetures information, it can easily fail to reflect on the true loss landscape from sampling viewpoint(see 5). For convex optimization view point, Hessian indeed captures the global information about the loss landscape, but for non-convex region, it can only express the local curvature information and sampling according to the local information can easily lead to sampling of high loss region.

**Direct training with different random seeds**. The posterior consist of models trained with different random seeds is studied in Fort et al. (2020) and shown be simple and strong base line for posterior for uncertainty measurement. The prediction of models trained with different random seeds provides strong diversity compared to the sampling methods which explore only the local region. In practice, it is easy to implement and applied to various to different leaning scenarios. The shortcoming of the method is that it requires much larger memory than other method as it stores and trains multiple models at the same time. Additionally, it did not necessarily violate the above observation as the models involved in the posteriors are trained simultanously and have low loss guarantee.

**Diagonal Gaussian** Diagonal Gaussian posteriors is well studied and shown to offer generalization guarantee. It provides easy control for the region to explore. Despite the fact that it did not contain local curvetures information, it can still fit in the observation from our theory as long as we select proper range for the variance. Compared to the two previously mentioned method, it offers better speed and memory consumption as we only need to sample independent variables for the construction of the whole distribution. In addition to the advantages mentioned, there is subtle connection between this posterior and optimization and we will illustrate in the following.

## A.6 Details about Greedy selection

Greedy selection method has been studied in many different prior works Khanna et al. (2017); Elenberg et al. (2017) to set up basis for its correctness and its applicability for different functions. As pointed out in Mirzasoleiman et al. (2020), one can transform the coreset problem on gradient

| Speed up | CIFAR10 | CIFAR100 | TinyImagenet |
|----------|---------|----------|--------------|
| Crest | 0.46122396 | 1.220357534 | 1.328621908 |
| Our | 1.71376899 | 1.227372798 | 1.448220165 |

Table 5: Our method obtain better speed up compare to the benchmark method and the results also generalized to different datasets and architectures. The speed up is calculated as (Full training time / Method training time). The results are average over 5 different random seeds.

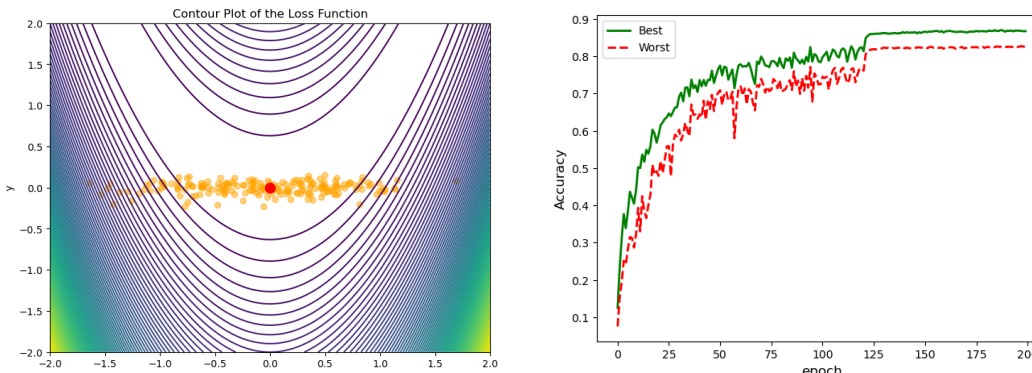

Figure 5: (Left) We calculate the Hessian for sampling at red dot and sampling (yellow dot) using the Hessian inverse as a covariance matrix. The motivation for performing sampling of this kind is that one expect model to be sample lying on the low loss region as it put larger probability density in the small eigenvalue direction. However, when the Hessian loss its ability to represent the true loss curvature (non-convex setting), the Hessian posterior can sample models in regions of high loss, even when the Hessian is computed correctly. (Right) For ensemble method, we find its performance is competitive to Gaussian posterior. However, we find that there exist performance divergence in the different models trained with different random seed and the coreset selected show different performance gain for different models. The coreset selected through this posterior may not be able offer best performance gain in the ensemble.

into the following submodular cover problem with constant $C$:

$$S^* = \operatorname*{argmin}_{S' \subseteq S} |S'| \ \text{ s.t } \ \sum_{j \in S'} \min_{i \in S} ||\nabla l_i(w_t) - \nabla l_j(w_t)|| \le \epsilon \tag{65}$$

The $\gamma$ in origin problem will be calculated as the number of times a specific sample $j \in S'$ is used to achieve minimum distance in the argument right hand side in this transformed problem. Greedy algorithm is used to calculate the sample being selected in which achieve time complexity $\mathcal{O}(nk)$ in existing work Mirzasoleiman et al. (2020); Killamsetty et al. (2021a); Pooladzandi et al. (2022), where $n$ is the size of the training set and $k$ is the number of samples being selected.

Despite linear complexity in terms of the sample selected, the calculation of the difference norm of the right hand side is still expansive due to the high dimensional properties of deep learning models. Several works Killamsetty et al. (2021b,a); Mirzasoleiman et al. (2020); Pooladzandi et al. (2022) demonstrate experimentally and theoretically that one can use the gradient with respect to the last layer for the calculation of gradient difference as it captures the norm of difference properly and greatly speed up the process for practical application, though a recent work has argued against it. Lastly, we complete the formulation with the formulation with fix sample size selected $k$ as following:

$$S^* = \operatorname*{argmax}_{S' \subseteq S} C - \sum_{j \in S'} \operatorname*{argmin}_{i \in S} ||\nabla l_i(w_t)^L - \nabla l_j(w_t)^L|| \tag{66}$$

$$\text{s.t } |S'| \le k$$

### A.7 Details about the algorithm design

In this section, we will briefly discuss the design of the our algorithm. First of all, instead of selecting the entire training set like Mirzasoleiman et al. (2020), we adapt from Yang et al. (2023) to select

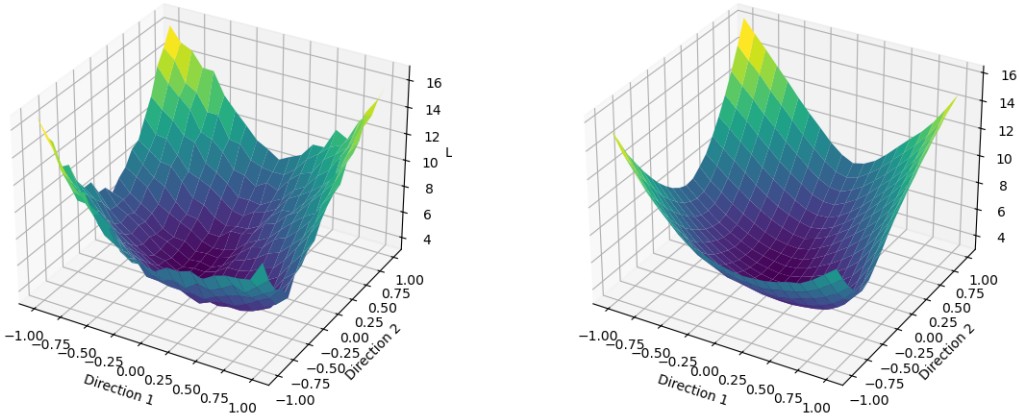

Figure 6: The loss landscape generated by coreset of Craig method and the coreset of our method. The graph is generated with CIFAR10 data and ResNet20 using 1% data budget.

from mini-batch and union the mini-batch to obtain coreset with specified size. This help to reduce the selection time as mentioned in the Yang et al. (2023). For the selection, we calculate the expected version as 6 to ensure the properties obtained in the theory remain true. We did not perform threshold check listed in the Yang et al. (2023), we instead perform update on each epoch.

### A.8 Toy model: Trajectory difference

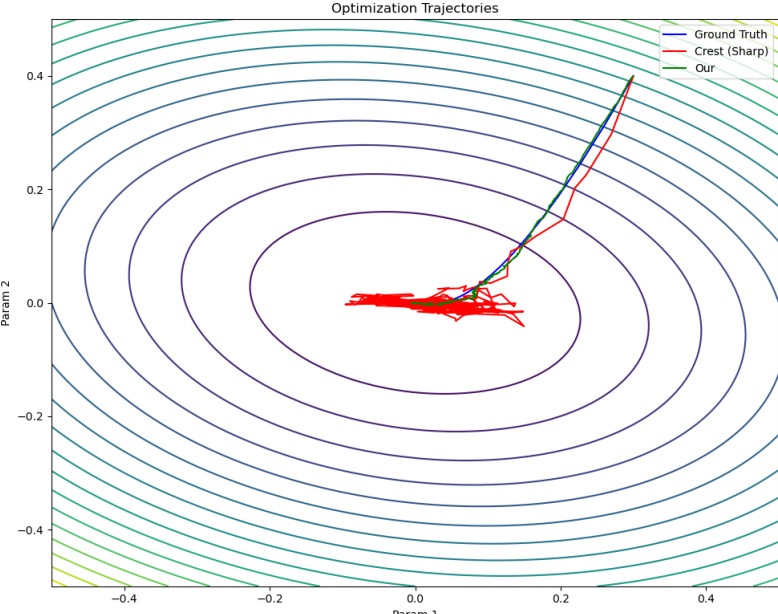

Figure 7: The trajectory difference between gradient descent, Crest and gradient descent. To mimic the gradient mismatch resulting from the label noise, we inject high frequency function into the Crest and our method. To mimic the smoothed version of the loss landscape, we reduce the magnitude of the gradient in our method. We can find that the injected noise can drastically change the trajectory of different methods while the smoothed version can help better recover the ground truth trajectory. The fluctuation from noise becomes more significant around minima as the gradient around minima becomes smaller under this simulated experiment.

## A.9 More learning scenarios. (Different training budget, corruption level, and architecture.)

In this section, we perform experiments on even more learning condition such as different corruption level, training budget, and architecture. For architecture, We extend our result to Vision transformer to identify whether or not attention based models can work properly with our method and the results show that our method already outperform others under various structures. For other learning setting, we also find our method consistently outperform other benchmark which further justify the robustness of our method.

| Budget | Corrupt Ratio | Our Method | Random | Crest |
|---|---|---|---|---|
| 0.2 | 0.0 | **0.8969 ± 0.0026** | 0.8968 ± 0.0026 | 0.8083 ± 0.0125 |
| | 0.1 | **0.8795 ± 0.0026** | 0.8788 ± 0.0032 | 0.8331 ± 0.0454 |
| | 0.3 | **0.8495 ± 0.0053** | 0.8483 ± 0.0027 | 0.7632 ± 0.0396 |
| | 0.5 | **0.8022 ± 0.0020** | 0.8011 ± 0.0032 | 0.5308 ± 0.0103 |
| 0.01 | 0.0 | **0.6683 ± 0.0044** | 0.6573±0.0211 | 0.4370 ± 0.0184 |
| | 0.1 | **0.6216 ± 0.0229** | 0.6147 ± 0.0272 | 0.3631 ± 0.0496 |
| | 0.3 | **0.5667 ± 0.0037** | 0.5431 ± 0.0272 | 0.3022 ± 0.0234 |
| | 0.5 | **0.4801 ± 0.0150** | 0.4412 ± 0.0172 | 0.1988 ± 0.0231 |

Table 6: CIFAR-10. Performance comparison of different methods across varying corruption ratios and budget levels. The best-performing method for each setting is highlighted in bold.

| Budget | Corrupt Ratio | Our Method | Random | Crest |
|---|---|---|---|---|
| 0.2 | 0.0 | **0.7337 ± 0.0013** | 0.7291 ± 0.0010 | 0.7172 ± 0.0029 |
| | 0.1 | **0.6712 ± 0.0022** | 0.6566 ± 0.0045 | 0.6357 ± 0.0033 |
| | 0.3 | **0.5521 ± 0.0021** | 0.5246 ± 0.0061 | 0.5081 ± 0.0064 |
| | 0.5 | **0.4535 ± 0.0014** | 0.3884 ± 0.0059 | 0.3500 ± 0.0094 |
| 0.01 | 0.0 | **0.2660 ± 0.0079** | 0.2521 ± 0.0039 | 0.1672 ± 0.0126 |
| | 0.1 | **0.2396 ± 0.0029** | 0.2231 ± 0.0033 | 0.1469 ± 0.0039 |
| | 0.3 | **0.1773 ± 0.0050** | 0.1566 ± 0.0025 | 0.1191 ± 0.0038 |
| | 0.5 | **0.1235 ± 0.0026** | 0.1134 ± 0.0051 | 0.0823 ± 0.0021 |

Table 7: CIFAR-100. Performance comparison of different methods with varying corruption ratios and budget levels. The best-performing method for each setting is highlighted in **bold**.

| Corrupt Ratio | 0.0 | 0.1 | 0.3 | 0.5 |
|---|---|---|---|---|
| Our Method | **0.8294 ± 0.0011** | **0.8077 ± 0.0008** | **0.7734 ± 0.0009** | **0.7064 ± 0.0026** |
| Crest | 0.8274 ± 0.0038 | 0.8002 ± 0.0042 | 0.7467 ± 0.0137 | 0.6653 ± 0.0112 |
| Random | 0.8200 ± 0.0034 | 0.7980 ± 0.0078 | 0.7479 ± 0.0040 | 0.6806 ± 0.0030 |

Table 8: Performance comparison of different selection methods using pretrained ViT-Base on CIFAR-100. Our method consistently outperforms both Crest and random sampling. We train with a learning rate of 0.0003, weight decay of 0.1, and a warm-up scheduling for 20 epochs. The full training process consists of 100 epochs to fit within the rebuttal time constraints.

| Method | 0.0 | 0.1 | 0.3 | 0.5 |
|---|---|---|---|---|
| Ours | 0.8773±0.0046 | 0.8893±0.0172 | 0.8787±0.0061 | 0.8767±0.0042 |
| Crest | 0.8707±0.0163 | 0.8727±0.0110 | 0.8593±0.0103 | 0.8727±0.0253 |
| Random | 0.7520±0.0040 | 0.7447±0.0142 | 0.7373±0.0186 | 0.7400±0.0243 |

Table 9: additional experiment on the TREC-50 language dataset using ELECTRA-small-discriminator, a compact transformer model with approximately 14 million parameters. Alongside SNLI (used in the main paper), this dataset represents a language inference-style task. We finetune the pretrained model for 50 epochs under a 10 percent data budget, using AdamW with a learning rate of 1e-4, weight decay of 0.01, and standard default settings. We also apply learning rate warm-up for 1 epoch to stabilize training.

| Corruption Ratio | RBFNN(Tukan et al., 2023) | Craig | CREST | Ours |
|---|---|---|---|---|
| 0.0 | 0.6449±0.0038 | 0.5665±0.0053 | 0.6728±0.0030 | 0.6986±0.0025 |
| 0.1 | 0.5248±0.0133 | 0.4629±0.0102 | 0.6391±0.0040 | 0.6644±0.0034 |
| 0.3 | 0.3422±0.0140 | 0.2968±0.0162 | 0.5712±0.0060 | 0.6085±0.0044 |
| 0.5 | 0.2868±0.0190 | 0.1603±0.0048 | 0.4305±0.0260 | 0.5014±0.0068 |

Table 10: CIFAR-100 test accuracy (mean ± std) under different corruption ratios. We compare our method with sensitivity based method (RBFNN(Tukan et al., 2023)) on CIFAR100 dataset under different corruption level.

| Corruption Ratio | RBFNN(Tukan et al., 2023) | Craig | CREST | Ours |
|---|---|---|---|---|
| 0.0 | 0.8255±0.0040 | 0.7618±0.0080 | 0.8724±0.0040 | 0.8757±0.0029 |
| 0.1 | 0.7924±0.0113 | 0.7490±0.0066 | 0.8440±0.0030 | 0.8544±0.0034 |
| 0.3 | 0.7280±0.0118 | 0.7050±0.0112 | 0.7843±0.0040 | 0.8120±0.0058 |
| 0.5 | 0.6216±0.0241 | 0.6320±0.0186 | 0.6652±0.0130 | 0.7318±0.0143 |

Table 11: CIFAR-10 test accuracy (mean ± std) under different corruption ratios. We compare our method with sensitivity based method (RBFNN(Tukan et al., 2023)) on CIFAR10 dataset under different corruption level.

# B Experiment details

## B.1 Code base

We develop our method base on code provided in Crest (Yang et al., 2023). For Craig and Glister, we use code based from CORD (`https://github.com/decile-team/cords`) with training hyperparameter changed to our setting.

## B.2 Datasets and architectures

In our work, we conduct experiments on various image datasets. MNIST (Deng, 2012), EMNIST (Cohen et al., 2017), CIFAR10, CIFAR100 (Krizhevsky, 2009), and Tinyimagenet (Russakovsky et al., 2015), SNLI (Bowman et al., 2015) and Imagenet-1k dataset. For MNIST and EMNIST datasets, we use Lenet. For CIFAR10, CIFAR100, Tinyimagenet and Imagenet-1k, we use respectively ResNet20, ResNet18 and ResNet50. For SNLI, we use pretrain RoBERTa (Liu et al., 2019) model. To creat data corruption, we pick specified portion of training samples and flip the corresponding label to other classes to ensure the corrupt ratio is rigorous. For all experiments except for Tinyimagenet, we run on single A10 GPU. For Tinyimagenet, we use single NVIDIA A100 GPU.

## B.3 Training hyperparameter

For all experiments, we fix the peak learning rate at 0.1 and total training epoch to 200. The batch size is set to 128. For the first 20 epochs, we use linear warm up until learning rate reach 0.1 and decrease the learning rate by factor 0.1 at 120 epoch and 170 epoch. These hyperparameters were consistent with those in Yang et al. (2023) and were chosen to ensure fair comparisons across methods and datasets.

## B.4 Experiment details about loss landscape and its matching

We generate the loss landscape plot using technics in Li et al. (2018). We purturb the model weights using

$$f(\alpha, \beta) = l(w^* + \alpha\delta + \beta\eta) \tag{67}$$

In which the $\delta$ and $\eta$ are two randomly initialized vectors with magnitude scaled to models parameters. The plots are generated using 20 by 20 grid and for each grid we calculate the loss on the whole training dataset, Craig subset, and our subset with the same parameter. In the plot, we incorporate our method to the Craig method and use Gaussian noise 0.01. We select all at once as Craig does to verify that the method will bring smoothness to the loss surface and we observe that there exist more sharp corner for the loss surface created by Craig and the loss surface generated by our method is smoother than Craig method. The loss for Craig and our method are scaled loss using the $\gamma$ constant obtain through the greedy selection subroutine. Both plot are generated using 1% data budget for each selection methods. The 3D plot is in Figure 6.

## B.5 Evaluation

We evaluated different methods on various datasets under different corruption ratios, with a fixed data budget of 10%. We recorded the final test accuracy and measured time as the process wall time (i.e., from the start to the end of the process). To ensure reproducibility, all experiments were conducted using 5 different random seeds, and results were averaged across these runs.

# C Time complexity analysis

For Craig, there exist the need for forward propogation for each sample, and the corresponding time complexity is $\mathcal{O}(dn)$. (d:model parameter, n:size of the dataset) The method is to select $q$ fraction ($0 < q < 1$) of the whole dataset with size all at once. The greedy selection strategy for selecting $qn$ samples out n samples is $\mathcal{O}(qn^2)$. Therefore, the complete time complexity is $\mathcal{O}(qn^2 + dn)$ for each epoch.

For Crest, instead of selecting from whole dataset, they select from subset with size M, and the corresponding time complexity for forward propogation is $\mathcal{O}(dR)$. (d:model parameter, n:size of the dataset) The method is to select $m$ point from the subset. The greedy selection strategy for selecting $Rm$ samples out n samples is $\mathcal{O}(Rm)$. Therefore, the time complexity is $\mathcal{O}(P(Rm + dR))$ where the $P$ is the number of subset that is manually chosen. The overall time complexity for Crest method need to multiple by the times they update the coreset within one epoch for fair comparison, but the average number of update for coreset cannot be calculated as they utilize adaptive strategy which depend on the curveture of the loss landscape.

For our method, we also select from subset with size R but we need multiple forward propogation (M times) and the corresponding time complexity is $\mathcal{O}(MdR)$. We then select $m$ points from the subset and the corresponding time complexity is $\mathcal{O}(mR)$. We need to perform multiple time to have $q$ portion of the whole dataset. The overall time complexity is $\mathcal{O}(q\frac{n}{R}(MdR + mR))$ for one epoch.

For CIFAR-10 dataset, for each epoch, the running time for forward pass to obtain the gradient is 0.136 second for each batch, and the running time for the greedy selection is 0.000612 second for each batch. Hence, a much larger time is spent on the forward pass instead of the greedy selection part of the algorithm. The current state of the art method CREST requires expensive tracking of the Hessian which results in significantly longer training time and memory footprint. As a result, our method remains competitive in terms of the total training time while offering improved selection quality.

# D    Assumptions in the theory

**Theorem 4.2 relies on assumptions of third-order smoothness and Hessian symmetry of the loss function**

The assumptions of third-order smoothness and Hessian symmetry are commonly used in deep learning theory to facilitate theoretical analysis. One key observation in deep learning is that the loss landscape often exhibits a degree of continuity, meaning that small changes in the parameter space generally lead to gradual changes in the loss function. This aligns with empirical findings on neural network optimization, where sharp transitions in loss are rare under typical training conditions.

The symmetry of the Hessian follows naturally from the continuity and differentiability of the loss function. There are several important results derived using these assumptions. [A, B, C, D, E, F, G](Martens, 2010; Kiros, 2013; Ghorbani et al., 2019; Kunin et al., 2021; Barshan et al., 2020; Yao et al., 2020; Bottou et al., 2018) While non-linear activation functions introduce complexities, prior works suggest that, in practice, the loss function remains smooth enough for such assumptions to be reasonable.[C, H] (Ghorbani et al., 2019; Liu et al., 2023). Additionally, there are lines of research trying to approximate the Hessian using Fisher Information matrix (FIM) such as (Pascanu & Bengio, 2014; Liao et al., 2018; Sen et al., 2024). This implicitly assume that the Hessian is symmetric as FIM is symmetric according to its definition. Also, there are works(Kirkpatrick et al., 2017; Ritter et al., 2018b) using Hessain as a precision matrix in probabilistic models, which implicitly assume symmetry in its structure and receive success in capturing or improving the behavior of deep learning.

Similarly, the third-order smoothness assumption extends this notion by ensuring that second-order derivatives do not change abruptly, which aligns with empirical observations about the optimization dynamics of deep networks. These smoothness and regularity conditions are standard in optimization theory((Jin et al., 2017a; Allen-Zhu & Li, 2018; Carmon et al., 2017; Criscitiello & Boumal, 2021; Jin et al., 2017b; Bottou et al., 2018)) and are widely used to analyze generalization and convergence properties of deep learning models.

Thus, while these assumptions may not hold universally in all settings (and we are not aware of any assumptions that hold universally for all models), they are reasonable approximations that enable theoretical insights into the learning dynamics of deep neural networks. We hope the reviewer agrees that our results are novel and useful within the context of current understanding of deep neural networks.

# E    Adacore comparison

AdaCore is related to our work in terms of its motivation to capture loss curvature. However, we were unable to include it in our experiments due to the lack of an accessible or functional implementation. We explored multiple sources, including the official AdaCore repository (https://github.com/opooladz/AdaCore.git), which has always been empty ever since it was created, as well as public coreset libraries such as CORD (https://github.com/decile-team/cords) and DeepCore (https://github.com/PatrickZH/DeepCore.git). We did not find a working implementation of AdaCore in any of these repositories.

We also attempted to reimplement the method based on the paper, but were unable to reproduce the reported performance. The method requires several manually tuned hyperparameters and includes steps involving Hessian computation, which are both computationally intensive and memory demanding. This introduces a significant runtime and scalability barrier, particularly problematic for the large-scale or noisy settings we focus on, and undermines the motivation for using coresets to speed up training.

Additionally, we note that AdaCore has not been included in recent coreset benchmarks, such as those by (Yang et al., 2023; Okanovic et al., 2023) which includes the authors of Adacore themselves, where efficiency and scalability are prioritized. We believe this omission reflects a broader consensus that AdaCore, while conceptually interesting, is not competitive in practice under modern resource constraints.

