# OpenReview forum: "Stable Coresets via Posterior Sampling: Aligning Induced and Full Loss Landscapes"
_NeurIPS.cc/2025/Conference — NeurIPS 2025 poster_

### Official Review · Reviewer_Qgh2 · 2025-06-24

**Clarity:** 2
**Significance:** 3
**Originality:** 3
**Rating:** 5
**Confidence:** 3

**Summary:**

The paper provides a methodology for submodular coreset generation that leverages gradients such that the loss of training on the coreset has lower discrepancy towards the loss of training on the entire dataset than previous submodular coreset generation methodologies. This is done by establishing a connection between the posterior sampling and loss landscapes. To that end, the authors then provide a loss function that takes into account a smoothed posterior sampling as a function of the model's weights.

Finally, experiments are provided showcasing the caveats of using previous submodular coreset generation that leverages gradients when handling corrupted data as well as the robustness of the provided coreset generation technique in the paper.

**Questions:**

Please address the following questions and edits:

1) Page 2, Line 71: replace "an" with "a".
2) Page 4, Algorithm 1: Wouldn't it be more correct to ensure that the union of $V_p$ would be equal to  $S$?
3) Page 5, Line 166: Remove second "instead".
4) Page 6, Theorem 3.3: What is $\sigma_2$?
5) Page 8, Line 264: Add "." at the end of the paragraph name.
6) Supplementary, Page 22: What is $w^\ast$?
7) Supplementary, Page 22, equation 15, last equality: There is a missing p(w).
8) Supplementary, Page 22, equation 15, last inequality: How was the negative term obtained?
9) Supplementary, Page 23, equation 20: How was the first term obtained?
10) Please ensure to update the proofs and ensure that every pass is justified correctly, since the theoretical contribution of the paper revolves around these proofs.
11) Why did the authors not compare against GradMatch?
12) How does your approach fare against sensitivity-based coreset, e.g., [1]? I am asking since this approach outperformed that of GradMatch.

---------------------------------
[1] Tukan, M., Zhou, S., Maalouf, A., Rus, D., Braverman, V., & Feldman, D. (2023, July). Provable data subset selection for efficient neural networks training. In International Conference on Machine Learning (pp. 34533-34555). PMLR.

**Ethical Concerns:**

["NO or VERY MINOR ethics concerns only"]

**Final Justification:**

I have increased my score as a result of the rebuttal and discussion with the authors here. The authors have addressed the lingering concerns I had with their submission, and as a result, the score has been updated.

**Limitations:**

Yes.

**Quality:**

3

**Strengths And Weaknesses:**

In what follows, the strengths of the paper are given:
  * The idea of leveraging gradients and ensuring lower discrepancy towards the gradients of the full model during the training process is novel and of high interest in this field.
  * The idea to use the BN layer for the gradients used for the coreset construction is smart and led to lower memory.
  * The experimental settings and practical results showcase the importance of the approach and its effect on the quality of the trained model.

To this end, below are the weaknesses of the paper:
   * The writing needs some polishing, as it is hard to understand some of the things in the paper -- mainly concerning the proofs (see questions section)

---

> ### Author Rebuttal · Authors · 2025-07-31
>
> 1. Page 4, Algorithm 1: Wouldn't it be more correct to ensure that the union $V_p$ of would be equal to $S$?
>
> We thank the reviewer for raising this point. It is not necessary in our setting for the union of $V_p$ to exactly equal $S$. For example, under a 10 percent data budget with a batch size of 128 and 100 total batches per epoch, we may set $P=10$ and construct each $V_p$ from $R = 500$ examples, selecting $m = 128$ samples from each for training. In this way, $S$ (the final coreset) is formed without requiring full dataset access. This design avoids the high computational cost and selection bias that may arise when scanning the full dataset, as in methods like Craig.
>
> 2. Page 6, Theorem 3.3: What is $\sigma_2$ ?
>
> We thank the reviewer for the question. In Theorem 3.3, $\sigma_2$ refers to the variance of the noise used to perturb the model weights as stated in assumption 4 in appendix A.3, while $\sigma_1$ denotes the variance of the stochastic gradient noise due to data sampling. Unlike $\sigma_1$, which arises inherently from minibatch training, $\sigma_2$ is a tunable parameter under our control and can be adjusted to optimize performance. We will clarify this distinction in the main text and reference the relevant derivations in the appendix for completeness.
>
> 3. Supplementary, Page 22: What is w*?
>
> We apologize for the typo. The $w*$ is indeed $\bar{w}$.
>
> 4. Supplementary, Page 22, equation 15, last inequality: How was the negative term obtained?
>
> we provide the detailed derivation for it below:
>
> \begin{equation}
> \begin{split}
>     f(\bar{w}) \int (f(w) - f(\bar{w}))p(w)dw &= f(\bar{w}) \int (f(w) - f(\bar{w}) -\nabla f(\bar{w})(w-\bar{w}) + \nabla f(\bar{w})(w-\bar{w}))p(w)dw \\
>     &= f(\bar{w}) \int ((f(w) - f(\bar{w}) -\nabla f(\bar{w})(w-\bar{w}))p(w)dw + f(\bar{w})\int \nabla f(\bar{w})(w-\bar{w}))p(w)dw \\
>     &= f(\bar{w}) \int ((f(w) - f(\bar{w}) -\nabla f(\bar{w})(w-\bar{w}))p(w)dw \\
>     &\leq ||f(\bar{w})|| \int ||(f(w) - f(\bar{w}) -\nabla f(\bar{w})(w-\bar{w})||p(w)dw \\
>     &\leq ||f(\bar{w})|| \int c_1||w-\bar{w}||p(w)dw \\
>     & = c_1 ||f(\bar{w})|| E_p[||w-\bar{w}||] \\
>     &\leq c_1 ||f(\bar{w})|| \sqrt{E_p||w-\bar{w}||^2} \\
>     &= c_1 ||f(\bar{w})|| \sigma \sqrt{d}
> \end{split}
> \end{equation}
>
> The second term at the second line of the derivation is canceled as $\int w p(w) dw = \bar{w}$ and the first inequality is due to the Cauchy-Schwarz inequality and the second inequality is due to the equation 14 in appendix A.1. The final inequality uses Jensen’s inequality due to the concavity of the square root function. Since we obtain the upper bound of the magnitude, we can lower bound the term by switching the sign of the term to be negative as claim in equation 15 and 16 in the appendix A.2. We acknowledge minor issues in the previous version (e.g., a missing constant or square root), but these do not affect the overall dependencies in the result. We will correct these technical details in the updated version.
>
> 5. Supplementary, Page 23, equation 20: How was the first term obtained?
>
> we provide the detailed derivation for it below:
>
> \begin{equation}
> \begin{split}
>     \int ||f(w) - f(\bar{w}) - \nabla f(\bar{w})(w - \bar{w})||^2 p(w)dw
>     &\geq \int c_2 ||w-\bar{w}||^2p(w)dw \\
>     &=c_2 E_p[||w-\bar{w}||^2] \\
>     &=c_2 \sigma^2d
> \end{split}
> \end{equation}
>
> The first inequality is due to the equation 14 in appendix A.2.
>
> 6. Please ensure to update the proofs and ensure that every pass is justified correctly, since the theoretical contribution of the paper revolves around these proofs.
>
> We thank the reviewer for raising the problem and help make the paper better. We have double checked the proofs and the result remains intact. We will update the proof in the updated version.
>
> 7. Why did the authors not compare against GradMatch?
>
> We thank the reviewer for raising this point. The short answer is that GradMatch is already outclassed by other methods we compare against e.g. CREST extensively evaluated against GradMatch and showed the former is superior.
>
> In designing our experimental comparison, we aimed to evaluate a diverse range of methods that reflect fundamentally different principles of subset selection. Specifically, we compared against: (1) \textbf{Craig}, which uses submodular greedy selection based on gradient similarity; (2) \textbf{Crest}, which adaptively selects points to match loss landscape curvature; (3) \textbf{Glister}, which employs bi-level optimization to maximize validation performance; and (4) random selection as a baseline. \textbf{GradMatch} is indeed another gradient matching method, but its core idea—greedy selection based on gradient alignment—is conceptually similar to Craig, with the main difference being the use of Orthogonal Matching Pursuit (OMP) rather than submodular cover. Since Craig already serves as a strong representative of this class, and other methods like Crest have shown strong empirical performance over both, we chose to focus our comparisons on methods with greater conceptual diversity. We will clarify this rationale in the updated version.
>
> 8. How does your approach fare against sensitivity-based coreset, e.g., [1]? I am asking since this approach outperformed that of GradMatch.
>
> We thank the reviewer for suggesting another baseline. The sensitivity-based baseline takes a more `global' perspective - constructing a non-uniform distribution to optimize over the worst case bound for an RBFNN on the original data. this in turn acts as a proxy for gradient matching, assuming the gradients can be closely approximated by an RBFNN. However, since the coreset is selected using worst case bounds to be able to fit /any/ RBFNN, it may not capture nuances of data distributions, especially in presence of noise. This is exactly what we observe too, as the following tables show. Note that we were not able to reproduce the results presented in their paper, may be due to different data setups even the random subsampling reported in that paper is way worse than what we observed, and what many other coreset papers have reported. (potentially due to update frequency, learning rate, architecture and other hyperparameters.) Nevertheless we use the code they made publicly available for our evaluations. We see that the method performs better than Craig but worse than crest which is worse than our method. This is especially apparent in the presence of noise.
>
> Cifar-100
>
> | Corruption Ratio | RBFNN [A]| Craig (mean ± std) | CREST (mean ± std) | Ours (mean ± std) |
> |------------------|------------|--------------------|---------------------|-------------------|
> | 0.0              |    0.6449  ± 0.0038       | 0.5665 ± 0.0053    | 0.6728 ± 0.0030     | **0.6986 ± 0.0025** |
> | 0.1              |    0.5248 ± 0.0133        | 0.4629 ± 0.0102    | 0.6391 ± 0.0040     | **0.6644 ± 0.0034** |
> | 0.3              |    0.3422 ± 0.014        | 0.2968 ± 0.0162    | 0.5712 ± 0.0060     | **0.6085 ± 0.0044** |
> | 0.5              |      0.2868 ± 0.019      | 0.1603 ± 0.0048    | 0.4305 ± 0.0260     | **0.5014 ± 0.0068** |
>
> Cifar-10
>
>
> | Corruption Ratio | RBFNN [A] | Craig (mean ± std) | CREST (mean ± std) | Ours (mean ± std) |
> |------------------|------------|--------------------|---------------------|-------------------|
> | 0.0              |   0.8255  ± 0.0040        | 0.7618 ± 0.0080    | 0.8724 ± 0.0040     | **0.8757 ± 0.0029** |
> | 0.1              |     0.7924 ± 0.0113       | 0.7490 ± 0.0066    | 0.8440 ± 0.0030     | **0.8544 ± 0.0034** |
> | 0.3              |     0.7280 ± 0.0118       | 0.7050 ± 0.0112    | 0.7843 ± 0.0040     | **0.8120 ± 0.0058** |
> | 0.5              |     0.6216 ± 0.0241       | 0.6320 ± 0.0186    | 0.6652 ± 0.0130     | **0.7318 ± 0.0143** |
>
>
> [A] Tukan, M., Zhou, S., Maalouf, A., Rus, D., Braverman, V., & Feldman, D. (2023, July). Provable data subset selection for efficient neural networks training. In International Conference on Machine Learning (pp. 34533-34555). PMLR.

---

> > ### Comment · Reviewer_Qgh2 · 2025-08-05
> > **Thank you**
> >
> > I would like to thank the authors for their clarifications and satisfactory responses.
> > I intend to increase my score to 5.

---

### Official Review · Reviewer_cZbS · 2025-06-26

**Clarity:** 2
**Significance:** 4
**Originality:** 4
**Rating:** 4
**Confidence:** 2

**Summary:**

The paper conjectures that existing coreset selection methods may not work well since they do not reflect the fact that the data may be noisy. Also, the paper focused the fact that, when we apply a coreset selection method (i.e., reducing the training set), the objective function of the training becomes less smooth. The paper focused on existing gradient-based coreset selection methods that selects a coreset so that the gradient of the objective function at the trained parameter is similar between the coreset and the full dataset. The proposed method altered this formulation that the gradient of the objective function "around" the trained parameter is similar between the coreset and the full dataset. They experimentally showed that this produces the objective function becomes smoother as the result of the coreset selection by the proposed method than by an existing method, and that the prediction performance was better than existing methods, especially the case when dataset is highly noisy.

**Questions:**

### Major comments

- Section 2, definition of $l$: Since the proofs of Theorem A.1 (Appendix A.2) and of Theorem 3.3 (Appendix A.3) require assumptions on $l$, assumptions on $l$ should be described in the main text. If it is not easy to describe in short, it may be good to provide the assumptions in the appendix and declare in the main text that the detailed assumption is not so restrictive but described in the appendix.
- Section 3, Overall: In this algorithm, if we take $\sigma\to 0$ does it imply the conventional method? I wondered to what extent the theoretical properties shown in this section holds for the conventional method.
- Section 3, Algorithm 1: Can the procedure of taking $E_\delta$ be described explicitly? Is it computed analytically or numerically? In addition, the paper states that "we focus on our attention for sampling in the batch normalization layers" (lines 178-179). How is it implemented in Algorithm 1?
- Section 3, subsection "Algorithm" (from line 158): Although theoretical analysis is done for the problem of equation (6) that finds the smallest coreset whose difference of gradients is $\epsilon$ or less, in reality Algorithm 1 selects a coreset of predetermined size whose difference of gradients is $\epsilon$ is smallest. In this sense, can the difference of gradients be represented by $M$, $R$ and/or $T$? In other words, can we derive the convergence analysis or other theoretical properties without $\epsilon$ (only with parameters controllable in Algorithm 1)?
- Section 3, Theorem 3.2: The statement (2) presents the difference between newton step of two subset as ${\cal O}(\epsilon^{\frac{1}{2}})$, but is it alright that we ignore $\sigma$ in this? If so, why?
- Section 3, Theorem 3.3: In the main text, the effect of $\xi_1$ (random sampling of batches) is not stated anywhere. As far as reading the appendix, does it mean that $\xi_1$ can be computed from $M$, $R$ and/or $T$? (If so, defining $\xi_1$ and $\xi_2$ as the variables with similar functionality is confusing; it should be clarified that $\xi_1$ is determined from other variables but we need to give assumptions on $\xi_2$.)
- Section 4, Overall: In the experiments $\sigma$ is selected via cross-validation, but what effect is expected if larger or smaller $\sigma$ is selected than the optimal one? Also, it is better if the effect of changing $\sigma$ is examined.
- Section 4, line 288: What is the "design of more effective posteriors"? As far as my understanding, it is not stated in Section 3 or Appendices. What is the mathematical formulation? How can it be implemented?

### Minor comments

- Section 2, lines 121-122: What does the sentence "For tractability and better generalization, in practice, we use SGD (stochastic gradient descent) for optimization of (1) can lead to large memory burden and can be inefficient due to the calculation time" intend? As long as I read this sentence, I understood that "we usually use SGD but it is inefficient", but it is somewhat unnatural.
- Section 3, equation (5): Should $|\ldots|_2^2$ be $\\|\\ldots\\|_2^2$ ?

### Formatting issues / Misspellings

- Overall: Figures and tables are too scattered in many pages even if they are related to experimental results. Please consider using `\begin{figure}[p]` and `\begin{figure}[p]` command: It creates pages that contain only figures and/or tables having `[p]`.
- Section 1, line 37: "Infact" --> "In fact"
- Section 3, line 146: "Shin et al. (2023)" is good to be replaced with the format "(Shin et al., 2023)" since it is not a citation directly read as a text.
- Section 3, line 205: "(see A.5)" --> "(see Appendix A.5)" is clearer.
- Section 4, Table 1: The place of Table 1 is unnatural. (If `\begin{figure}[p]` above is not applied,) please consider placing at the top of the page, or at the top or the end of the paragraph that refers the table.
- Section 4, line 279: "Training dyanamics" --> "Training dynamics"

**Ethical Concerns:**

["NO or VERY MINOR ethics concerns only"]

**Final Justification:**

First I rated 3 on this paper, but raised to 4 after rebuttals. My questions become clear, and the manuscript is expected to become clearer after fixes described in the rebuttal. Especially, for a parts of the paper with insufficient explicitness, I expect that these points will be improved in the final paper after reading authors' rebuttals.

**Limitations:**

The paper states that the limitation of the work is that they have examined performances for limited types of datasets (in this paper image datasets are used). I expect that, in the future work, not only the difference of data types but also the effect of model size should be examined.

**Paper Formatting Concerns:**

Nothing particular.

**Quality:**

2

**Strengths And Weaknesses:**

Strengths:

- Theoretical evaluations itself seems to be sound and useful.
- The algorithm is simple. Experimental results show that it is faster than existing methods.

Weaknesses:

- Several additional analyses are desired. Especially, the effect of $\sigma$ seems to be examined more (both theoretically and experimentally).
- Explanation of algorithms and experiments seem to be incomplete (see "Questions" section below for details).

---

> ### Author Rebuttal · Authors · 2025-07-30
>
> We thank the reviewer for all the comment and address them in the following. If there is still confusion, we will be glad to provide more information or illustration as needed in the discussion stage.
>
> 1. Section 2, definition of l: Since the proofs of Theorem A.1 (Appendix A.2) and of Theorem 3.3 (Appendix A.3) require assumptions on loss, assumptions on should be described in the main text. If it is not easy to describe in short, it may be good to provide the assumptions in the appendix and declare in the main text that the detailed assumption is not so restrictive but described in the appendix.
>
> We thank the reviewer for pointing this out. We will explain the assumption in main context or appendix to make it clear.
>
> 2. Section 3, Overall: In this algorithm, if we take $\sigma$ to zero does it imply the conventional method? I wondered to what extent the theoretical properties shown in this section holds for the conventional method.
>
> Yes, when $\sigma \to 0$, our method reduces to conventional deterministic approaches such as Craig and Crest. The gradient perturbations vanish, and the selection becomes equivalent to greedy optimization over exact gradients. Importantly, our theoretical results are consistent in this limit. Also, when $\sigma > 0$, our framework introduces loss landscape matching, which helps align coreset selection with the local geometry of the loss landscape. This provides a principled way to improve robustness over standard methods, offering both a generalization and a deeper interpretation.
>
> 3. Section 3, Algorithm 1: Can the procedure of taking expectation $E_\delta$ be described explicitly? Is it computed analytically or numerically? In addition, the paper states that "we focus on our attention for sampling in the batch normalization layers" (lines 178-179). How is it implemented in Algorithm 1?
>
> We compute the expectation over $\delta$ numerically by averaging gradients over 4 random perturbations (by sampling $\delta$ with gaussian distribution and add them to the parameter weight) of the model weights. For efficiency and stability, we restrict perturbations to batch normalization layers. This choice significantly reduces memory cost and avoids destabilizing the gradient scale, while still improving performance. Although perturbing all parameters is theoretically appealing, we found that focusing on batch norm layers offers a good trade-off in practice.
>
> 4. Section 3, subsection "Algorithm" (from line 158): Although theoretical analysis is done for the problem of equation (6) that finds the smallest coreset whose difference of gradients is $\epsilon$ or less, in reality Algorithm 1 selects a coreset of predetermined size whose difference of gradients is $\epsilon$ smallest. In this sense, can the difference of gradients be represented by M, R and/or T? In other words, can we derive the convergence analysis or other theoretical properties without (only with parameters controllable in Algorithm 1)?
>
> We thank the reviewer for this thoughtful question. The main theoretical tool we rely on is greedy selection for submodular cover problems, where the approximation guarantee is of the form $(1 - \exp(-\gamma))$, assuming $\gamma$-weak submodularity. However, the constant $\gamma$ depends on properties of the loss landscape and the feature geometry, which are data-dependent and not explicitly controlled by $M$, $R$, or $T$ in Algorithm 1. Therefore, this quantity cannot be determined explicitly through the control parameters presented in the algorithm and remain it in symbolic format in our analysis. Empirically, $\epsilon$ is small (see details in Craig paper) with greedy algorithm supporting the practical effectiveness of our fixed-size selection strategy. The theoretical gap between $\epsilon$-based guarantees and fixed-size selection is common in coreset literature (e.g., Craig and Crest), and we adopt a similar pragmatic approach.
>
> 5. Section 3, Theorem 3.2: The statement (2) presents the difference between newton step of two subset without dependency of $\sigma$, but is it alright that we ignore in this? If so, why?
>
> Thank you for the insightful question. While the simplified statement in the main text omits $\sigma$ for clarity, the full derivation in Appendix A.2 (e.g., Eq. 33) shows that certain terms do depend on $\sigma$, such as $(\frac{\epsilon}{\sigma})^{1/2}$. However, this dependency is entangled with other factors like the eigenvalues of the Hessian. We chose to highlight the dominant structural terms in the main text but agree it's helpful to clarify $\sigma$'s role. We will revise the paper to make this dependency more explicit.
>
> 6. Section 3, Theorem 3.3: In the main text, the effect of $\xi_1$ (random sampling of batches) is not stated anywhere. As far as reading the appendix, does it mean that $\xi_1$ can be computed from $M$, $R$ and/or $T$? (If so, defining $\xi_1$ and $\xi_2$ as the variables with similar functionality is confusing; it should be clarified that is determined from other variables but we need to give assumptions on $\xi_2$.)
>
> We appreciate reviewer for pointing this out. In our analysis, the structure of noise is assumed in equation in assumption part of the proof (see appendix A.2 and equation (49) (50) (57)) (The $\xi_1$ is isotropic noise with variance $\sigma_1$) as done in Crest.  We will add additional note for this in the main content to make it clear.
>
> 7. Section 4, Overall: In the experiments, $\sigma$ is selected via cross-validation, but what effect $\sigma$ is expected if larger or smaller is selected than the optimal one? Also, it is better if the effect of changing is examined.
>
> In our analysis, we find there exist trade off between matching of gradient and loss landscape. If we increase the magnitude of the noise injected into the model for selection, we will find that there exist better match in loss landscape for current coreset, and decrease of the noise will lead to more accurate estimation of the gradient at current model weight. The balance between the two gives the optimal performance. We also find if we increase the noise by too much, it will start to resemble the random sampling as the gradient becomes too chaotic and cannot capture either the loss landscape or gradient.
>
> 8. Section 4, line 288: What is the "design of more effective posteriors"? As far as my understanding, it is not stated in Section 3 or Appendices. What is the mathematical formulation? How can it be implemented?
>
> Thank you for pointing this out. The "design of more effective posteriors" refers to the flexibility in choosing how we perturb the model weights during gradient estimation. The current strategy can be interpreted as drawing samples from an isotropic Gaussian posterior centered at $\bar{w}$ with variance $\sigma^2$ (i.e., uniform in all directions). However, one could consider more informed posterior distributions. For instance, perturbing only along sharpest directions in the weight space—akin to using an anisotropic or structured posterior—may better explore sensitive regions of the loss landscape. While Section 4 outlines this idea at a high level, implementation details for our current perturbation strategy are provided in Appendix A.4. We agree that further formalization of alternative posteriors is an exciting direction for future work and will clarify this in the revision.
>
> 9. Section 2, lines 121-122: What does the sentence "For tractability and better generalization, in practice, we use SGD (stochastic gradient descent) for optimization of (1) can lead to large memory burden and can be inefficient due to the calculation time" intend? As long as I read this sentence, I understood that "we usually use SGD but it is inefficient", but it is somewhat unnatural.
>
> We appreciate that the reviewer pointing this out. There should be gradient descent (GD) in the front for comparison.

---

> > ### Comment · Reviewer_cZbS · 2025-08-05
> >
> > Thank you for responses. My questions become clear. Especially, for parts with insufficient explicitness, reflecting these responses will make the paper clearer.

---

> > > ### Author Response · Authors · 2025-08-05
> > > **Thank you for your comment**
> > >
> > > Thank you for your response — we're very glad to hear that your concerns have been addressed. We also appreciate your detailed review, and we will incorporate the feedback in the final version, it will  make our paper stronger. if you feel that after the rebuttal our paper meets a higher standard, we’d greatly appreciate it if you could reflect that by increasing your rating. Your updated evaluation would mean a lot to us in this competitive process.

---

### Official Review · Reviewer_qG1y · 2025-06-27

**Clarity:** 3
**Significance:** 2
**Originality:** 2
**Rating:** 4
**Confidence:** 3

**Summary:**

This paper proposes a new framework for coreset selection. A method for choosing representative data subsets to speed up deep learning training. The authors address challenges in existing gradient-based methods, such as strong SGD baselines and issues with representativeness due to changing loss curvature. Their approach builds a connection between posterior sampling and the loss landscape, and introduces a smoothed loss function to improve stability and generalization. They also provide a new convergence analysis and show through experiments that their method outperforms existing techniques in both training speed and generalization.

**Questions:**

1. Can you provide results on one more language or multimodel dataset on small models? It would make the result more concrete.
2. Can you theoretically compute time complexity of crest, craig and your method? (I have seen your method's complexity on appendix C but also want to see the difference and comparison of others.) If possible, please provide system runtime for training as well.

**Ethical Concerns:**

["NO or VERY MINOR ethics concerns only"]

**Final Justification:**

I would like to keep the original positive score since the authors solved most of my concerns.

**Limitations:**

1. This work has not been tested on larger models and datasets, which weakens its generality.

2. The paper does not report actual training time and only provides theoretical and data usage analysis. I would encourage the authors to include a more comprehensive evaluation of system performance.

**Quality:**

3

**Strengths And Weaknesses:**

Strength:

1. Amazing efficiency/accuracy improvement on training.

2. Very comprehensive ablation study about corruption ratio and accuracy.

Weakness:

1. Does not prove effectiveness on scalability especially on multimodality and language tasks.

2. Can not prove it is efficienct(in throughputs/latency) in real system.

---

> ### Author Rebuttal · Authors · 2025-07-30
>
> We appricate the reviewer comment and address them in the following:
>
> 1. Can you provide results on one more language or multimodel dataset on small models? It would make the result more concrete.
>
> We thank the reviewer for the suggestion. To strengthen our claims, we conducted an additional experiment on the TREC-50 language dataset using ELECTRA-small-discriminator, a compact transformer model with approximately 14 million parameters. Alongside SNLI (used in the main paper), this dataset represents a language inference-style task. We finetune the pretrained model for 50 epochs under a 10 percent data budget, using AdamW with a learning rate of 1e-4, weight decay of 0.01, and standard default settings. We also apply learning rate warm-up for 1 epoch to stabilize training. The results below show that our method outperforms both Crest and random selection under various corruption level:
>
> | Method  | 0.0           | 0.1           | 0.3           | 0.5           |
> |---------|---------------|---------------|---------------|---------------|
> | Ours    | 0.8773±0.0046 | 0.8893±0.0172 | 0.8787±0.0061 | 0.8767±0.0042 |
> | Crest   | 0.8707±0.0163 | 0.8727±0.0110 | 0.8593±0.0103 | 0.8727±0.0253 |
> | Random  | 0.7520±0.0040 | 0.7447±0.0142 | 0.7373±0.0186 | 0.7400±0.0243 |
>
> 2. Can you theoretically compute time complexity of Crest, Craig and your method? (I have seen your method's complexity on appendix C but also want to see the difference and comparison of others.) If possible, please provide system runtime for training as well.
>
> We thank the reviewer for asking the question. The main analysis of time complexity lies on the selection of subset and we provide comprehensive comparison here.
>
> For Craig, there exist the need for forward propogation for each sample, and the corresponding time complexity is $\mathcal{O}(dn)$. (d:model parameter, n:size of the dataset) The method is to select $q$ fraction ($0 < q < 1$) of the whole dataset with size all at once. The greedy selection strategy for selecting $qn$ samples out n samples is $\mathcal{O}(qn^2)$. Therefore, the complete time complexity is $\mathcal{O}(qn^2 + dn)$ for each epoch.
>
> For Crest, instead of selecting from whole dataset, they select from subset with size M, and the corresponding time complexity for forward propogation is $\mathcal{O}(dR)$. (d:model parameter, n:size of the dataset) The method is to select $m$ point from the subset. The greedy selection strategy for selecting $Rm$ samples out n samples is $\mathcal{O}(Rm)$. Therefore, the time complexity is $\mathcal{O}( P (Rm+ dR))$ where the $P$ is the number of subset that is manually chosen. The overall time complexity for Crest method need to multiple by the times they update the coreset within one epoch for fair comparison, but the average number of update for coreset cannot be calculated as they utilize adaptive strategy which depend on the curveture of the loss landscape.
>
> For our method, we also select from subset with size R but we need multiple forward propogation (M times) and the corresponding time complexity is $\mathcal{O}(MdR)$. We then select $m$ points from the subset and the corresponding time complexity is $\mathcal{O}(mR)$. We need to perform multiple time to have $q$ portion of the whole dataset. The overall time complexity is $\mathcal{O}(q\frac{n}{R} (MdR+mR))$ for one epoch.
>
> 3. The paper does not report actual training time and only provides theoretical and data usage analysis. I would encourage the authors to include a more comprehensive evaluation of system performance.
>
> We appreciate the reviewer’s suggestion. We acknowledge the importance of reporting practical system performance in addition to theoretical analysis. We have already included some relevant metrics: Memory usage across the entire training process is presented in Figure 2. Training time for different datasets is provided in Appendix A.5, Table 5. A brief comparison of training time between Crest and our method is discussed in the caption of Table 1 in the main paper. In the updated version, we will include a dedicated table that summarizes training time, memory usage, and coreset selection overhead across methods for a more comprehensive system-level evaluation.
>
> 4. This work has not been tested on larger models and datasets, which weakens its generality. The paper does not report actual training time and only provides theoretical and data usage analysis. I would encourage the authors to include a more comprehensive evaluation of system performance.
>
> We thank the reviewer for raising this concern. To address the issue of generality, we would like to clarify that we have conducted extensive evaluations across a wide range of datasets and model sizes: Datasets: Our experiments include MNIST, CIFAR-10, CIFAR-100, TinyImageNet, ImageNet, and SNLI (a language inference dataset), with dataset sizes ranging from 60,000 to 1.4 million samples. Model scales: We evaluate models from small to large, including LeNet, ResNet-20, ResNet-18, ResNet-50, RoBERTa, and ViT-Base, spanning parameter counts from 10K to over 100M. (See Appendix A.9 for ViT results.) Robustness checks: We test under varying data corruption ratios (0 to 90 percent), data budgets (1, 10, 20 percent), and 3–5 random seeds to ensure consistent performance. As for system performance, we report memory usage in Figure 2 and include training time comparison in Appendix A.5, Table 5. A more detailed runtime and memory comparison table will be added in the updated version for completeness. To our knowledge, little prior work on coreset selection has systematically evaluated across such a broad combination of model sizes, dataset types, corruption levels, and data budgets. We believe this provides a comprehensive demonstration of the method’s robustness and scalability.

---

> > ### Comment · Reviewer_qG1y · 2025-08-03
> >
> > The authors solved my concerns. I would like to keep the orignal score.

---

### Official Review · Reviewer_Pvnw · 2025-07-02

**Clarity:** 3
**Significance:** 3
**Originality:** 3
**Rating:** 5
**Confidence:** 3

**Summary:**

This paper proposes a new coreset selection method: using posterior smoothing for coreset selection.  The idea is to sample model weight perturbations from a Gaussian posterior centered at the current parameters, and use these perturbed weights to estimate the landscape and evaluate the coreset selection criteria.  The proposed approach essentially smooths out the loss landscape by averaging weights in a local neighborhood. The approach is gradient-based, and doesn’t need to compute the Hessians.

This paper provides convergence analysis and empirical validation for the proposed approach.

**Questions:**

Have any experiments been performed on transformer based models?

**Ethical Concerns:**

["NO or VERY MINOR ethics concerns only"]

**Final Justification:**

This paper proposes a new coreset selection method. The proposed posterior sampling method is intuitive and solid. My questions were adequately answered in the rebuttal. Feedback from other reviewers is also generally positive. I would like to maintain my rating for acceptance.

**Limitations:**

Yes.

**Paper Formatting Concerns:**

No formatting issues.

**Quality:**

3

**Strengths And Weaknesses:**

The paper is generally well written. Coreset selection is an interesting and important topic when training with a large amount of data. It is also useful for continual learning. The proposed posterior sampling method is intuitive and solid. The paper also provides convergence analysis for the proposed approach.
Experiments were conducted on various datasets and model architectures to validate the approach.  The model architectures used include various sizes of ResNet and LeNet.

The paper results would be stronger if it has results on transformer based model architectures.

---

> ### Author Rebuttal · Authors · 2025-07-30
>
> We appreciate Reviewer for the comment and summarize the quesions and responses in the following:
>
> 1. Have any experiments been performed on transformer based models?
>
> Yes. As shown in Appendix A.9, Table 8, we conduct experiments using ViT-Base on the CIFAR-100 dataset, and our method consistently outperforms other baselines across different corruption ratios. Additionally, we evaluate our method on transformer-based language models, specifically RoBERTa (a variant of BERT utilizing attention-based mechanisms), on the SNLI natural language inference dataset. (in the main context) In both the vision and language domains, our approach demonstrates improved performance over standard baselines, suggesting its general applicability to transformer architectures.

---

> > ### Comment · Reviewer_Pvnw · 2025-08-05
> >
> > Thanks for the rebuttal response. My questions are adequately answered. I would like to maintain my rating for acceptance.

---

### Comment · Area_Chair_KvXm · 2025-08-04
**rebuttal acknowledgement and discussion**

Dear reviewers,

Please make sure you acknowledge that you have read the authors' rebuttal, which is mandatory this year.  Also, please discuss with the authors if you have questions or comments on their rebuttal.  For those have done so, thank you!

Thanks,

AC

---

### Note · Authors · 2025-08-14

We appreciate everyone for engaging in the discussion to make the work better. We note that the reviewers were positive about algorithmic, theoretical as well as empirical contributions with concerns all addressed and no lingering problem. Here, we provide a brief summary of the reviewers' comments and our responses to help understand our work:
1. Reviewer Pvnw:
Asked us if we did any Transformer experiments for which we pointed the reviewer to our results in the appendix where we showed superiority of our methods in transformer architectures, such as ViT and RoBERTa, to ensure the generalality of our method. Reviewer Pvnw responded saying their questions are addressed and maintained the rating of 5 as the final score.
2. Reviewer qG1y:
Asked for more experiments for one more language dataset, and time complexity analysis for our method vs baselines. In our response, we provided additional results on the TREC-50 language dataset using the ELECTRA-small-discriminator, which demonstrated strong improvement compared to other methods. We also provided complexity analysis to compare different benchmarks. Lastly, we provided an explanation regarding the robustness of our methods by detailing the experimental setup and scale of the experiments to help address the issue. Finally, the reviewer agreed that the concerns were resolved and will keep the original rating of 4.
3. Reviewer cZbS:
Asked for clarifications about the algorithm and the relationship between variables used throughout the analysis, such as the role of noise injected and whether the error can be described through the controllable variables in the algorithm. In the response, we provided a detailed answer to these questions. The reviewer responded saying that their doubts problems were resolved. The reviewer's questions helped us improve the clarity in the paper, and we will incorporate their suggestions into the paper to improve exposition.
4. Reviewer Qgh2:
Had doubts in the proof and asked for the comparison with another benchmark called RBFNN, which is an intrinsically sensitivity-based algorithm. In our response, we provided a detailed derivations to clarify the steps in the proof. Also, we compared with the benchmark required by the reviewer and demonstrated that our method still holds a strong advantage over the sensitivity-based selection method. **Finally, the reviewer agreed that concerns were resolved and will raise the score to 5 (even though we still see it to be 4 for some reason?**)

---

### Decision · Program_Chairs · 2025-09-17

**Decision:**

Accept (poster)

**Comment:**

This paper proposes an approach based on posterior sampling for coreset selection. This approach introduces a smoothed loss function on model weights to improve the stability and generalization. The authors also analyze its convergence behavior and give the convergence rate. Extensive experiments are carried out on a broad variety of datasets and model architectures of various sizes. The authors show that the proposed coreset selection algorithm outperforms some of the SOTA methods. All reviewers consider the work sufficiently novel with a good impact to the machine learning community. There are numerous concerns raised by the reviewers including additional evaluation on more datasets/models and detailed questions on mathematical treatment. The authors' rebuttal has cleared up these concerns. Overall, this is an interesting work that is theoretically sound and practically well evaluated. All reviewers are supportive of accepting it.  The authors need to improve the exposition to make the paper clearer and more rigorous in the revised version.